# Astroglial exosome HepaCAM signaling and ApoE antagonization coordinates early postnatal cortical pyramidal neuronal axon growth and dendritic spine formation

Shijie Jin [1], Xuan Chen[1], Yang Tian [1], Rachel Jarvis[1], Vanessa Promes[1] & Yongjie Yang [1,2] ✉

Developing astroglia play important roles in regulating synaptogenesis through secreted and contact signals. Whether they regulate postnatal axon growth is unknown. By selectively isolating exosomes using size-exclusion chromatography (SEC) and employing cell-type specific exosome reporter mice, our current results define a secreted astroglial exosome pathway that can spread long-range in vivo and stimulate axon growth of cortical pyramidal neurons. Subsequent biochemical and genetic studies found that surface expression of glial HepaCAM protein essentially and sufficiently mediates the axon-stimulating effect of astroglial exosomes. Interestingly, apolipoprotein E (ApoE), a major astroglia-secreted cholesterol carrier to promote synapto-genesis, strongly inhibits the stimulatory effect of astroglial exosomes on axon growth. Developmental ApoE deficiency also significantly reduces spine density of cortical pyramidal neurons. Together, our study suggests a surface contact mechanism of astroglial exosomes in regulating axon growth and its antagonization by ApoE, which collectively coordinates early postnatal pyramidal neuronal axon growth and dendritic spine formation.

Developmental neuronal axon outgrowth and synaptogenesis are crucial steps in forming sophisticated and functional connectivity in the mammalian central nervous system (CNS). It is well established that synaptogenesis begins after birth and continues for several weeks postnatally while axon outgrowth is mostly completed at birth[1]. However, descending corticospinal tract (CST) axons that predominantly originate from layer V pyramidal neurons of the primary motor cortex continue to grow and reach spinal cord segments from postnatal day 1 to 10 (P1 to P10) in mice[2]. Developing astroglia have been well demonstrated to actively promote synaptogenesis and synapse maturation[3]. Early studies established that glia-derived cholesterol, transported by the astroglia-secreted lipoprotein ApoE, serves as a robust synaptogenic factor for retinal ganglion cells (RGCs)[4].

Several other secreted proteins from astroglia, such as Thrombos-pondin 1 and 2 (Tsp1/2)[5], Hevin[6], glypicans[7], and Chordin-like 1[8], have been later identified to promote excitatory synapse formation and stimulate glutamatergic activity. In contrast, far less is understood about whether and how developing astroglia regulate axon growth.

Developmental axon growth is driven by the actin and microtube dynamics within axonal growth cones as a result of receptor activation by extracellular trophic factors, adhesion molecules, and matrix proteins[9]. Although many of these ECM/adhesion proteins, such as neural cell adhesion molecule (NCAM), N-cadherin, and integrins, are highly expressed in developing neurons and neural progenitors[10], transcriptome profiling has found expression of a number of ECM and CAM genes in developing astroglia[11]. Early studies showed that γ-

[1]Department of Neuroscience, Tufts University School of Medicine, Boston, MA 02111, USA. [2]Graduate School of Biomedical Sciences, Tufts University, Boston, MA 02111, USA. ✉e-mail: yongjie.yang@tufts.edu

protocadherins (γ-Pcdhs) are also expressed by astroglia which promotes synaptogenesis in vitro and in vivo[12]. Genetic studies found that astroglial expression of neuroligins is important for developmental astroglial morphogenesis[13]. Neuronal cell adhesion molecule (NrCAM) was also found at astroglial process to regulate astroglia-inhibitory synapse interaction[14]. In particular, hepatocyte cell adhesion molecule (HepaCAM, also known as GlialCAM), a CAM protein containing immunoglobulin (Ig)-like extracellular domains[15], is highly enriched in (astro) glia in the CNS[16]. HepaCAM has been identified as a binding partner for a voltage-gated chloride channel Clc-2 and its mutations have been implicated in causing a rare form of leukodystrophy[17,18]. HepaCAM was also recently shown to regulate astroglial domain territory and gap junction coupling[19]. Whether astroglial CAM proteins including HepaCAM play a role in developmental axon growth remains unknown.

Exosomes (50–150 nm in diameter), a major type of secreted extracellular vesicles (EVs), are derived from intraluminal vesicles (ILVs) in the early endosomal compartment and are released from multivesicular bodies (MVBs) during endosome maturation[20]. EVs and exosomes secreted from various CNS cell types have been shown to regulate activity-dependent translation[21] and glutamate transporter function[22], to promote axon myelination and transport[23], and to maintain brain vascular integrity[24]. Whether astroglial exosome signals play a role in regulating neuronal functions has just begun to be understood. Astroglia-derived extracellular vesicles (ADEVs) are able to modulate dendritic complexity of cultured hippocampal neurons[25]. An extracellular matrix protein, fibulin-2, was also recently identified as astrocyte EV cargo that promotes synapse formation in a TGFβ-dependent manner[26]. However, the ultracentrifugation (UC) approach used in these studies to isolate astroglial exosomes often leads to mixed exosomes and secreted proteins[27], potentially undermining the effects mediated by astroglia secreted exosomes.

In the current study, we investigated the developmental function of astroglial exosomes, especially surface HepaCAM signaling, in regulating axon growth of cortical pyramidal neurons and how this pathway is antagonized by ApoE, which collectively coordinates early postnatal pyramidal neuronal axon growth and dendritic spine formation.

## Results

### Size exclusion chromatography (SEC)-isolated astroglial exosomes (A-Exo.) stimulate axon growth of cortical neurons

Exosomes have been conventionally isolated from cell culture medium or body fluids using serial (ultra)centrifugation steps[28]. However, recent studies have shown that UC-isolated exosomes are often contaminated with secreted proteins from cells[27,29]. We initially isolated A-Exo. from astrocyte conditioned medium (ACM, conditioned from >90% confluent astrocytes for 3 days) using the UC method and detected well-validated exosome markers, including tetraspanin family proteins CD63/CD81 and the ESCRT protein Tsg101[20], together with several astroglia-secreted proteins, such as Tsp1/2, Hevin, Sema3A, and Sparc (Supplementary Fig. 1a) that were previously identified as astroglia-secreted synaptogenesis modulators[3]. To better separate A-Exo. from secreted proteins, we optimized exosome isolation procedures using filtration (0.22 μm) and SEC (Fig. 1a). Immunoblotting of astroglia-secreted proteins and exosome markers in representative eluted fractions from ACM showed that Tsp1/2, Sema3A, Hevin, and Sparc are only detected in exosome-free but not in CD81+ A-Exo. fractions of ACM (Fig. 1b). Additional exosome markers CD63 and Alix were also selectively detected in exosome fractions of ACM (Fig. 1b). In addition, we found no contamination of typical subcellular organelles in A-Exo. indicated by undetectable Calreticulin, GM130, and Histone H2A, nor cytoskeleton proteins GFAP and β-actin (Supplementary Fig. 1b). On the other hand, glutamate transporter GLAST protein (mostly monomers) were detected in A-Exo. (Supplementary Fig. 1b). Subsequent immunoEM analysis of CD63 in eluted fractions further confirmed that CD63+ exosomal vesicles are detected only in exosome (#7–8) and mixed (#9) fractions (white arrows, Fig. 1c ii, iii) but not in other ACM fractions (Fig. 1c i, iv and Supplementary Fig. 1c). Notably, translucent CD63⁻ small vesicles (30–40 nm size range), possibly exomeres[30], were observed in certain eluted fractions, especially in exosome-free ACM fractions (yellow arrows, Fig. 1c ii, iii and Supplementary Fig. 1c). Exosome fractions were also analyzed by the qNano particle analyzer[31] from which a single Gaussian peak at a mean of 70–80 nm (Supplementary Fig. 1d) was revealed, confirming the population of nanovesicles with the typical size of exosomes but not microvesicles that are often shredded from plasma membrane and tend to be larger in size (100~1000 nm).

Whether and how A-Exo. influence neuronal properties is little known. Although tetraspanin protein (CD63 or CD81) immunoprecipitation (IP) can selectively isolate A-Exo., removing exosomes from IP beads has been difficult and the wash solution often kills neurons. As an alternative, we directly treated cultured cortical neurons with SEC-eluted pre-exosome (#4–6), exosome (#7–8), and post-exosome (#10–12 and #19–21, respectively) fractions from ACM for 24 h. Interestingly, βIII-tubulin+ neurites from neuronal cultures treated (at DIV 4) with exosome fractions, but not other fractions, are substantially longer than in the untreated control (Supplementary Fig. 2a, b). A-Exo.-stimulated neurite growth is also treatment time-dependent, with <10% or > 50% of neurites longer than 600 μm after either 1 or 3 days treatment, respectively (Supplementary Fig. 2c). In contrast, HEK cell-secreted exosomes have no stimulating effect on neurite growth (Supplementary Fig. 2d), indicating a specific effect of A-Exo. on neurite growth. As we observed CD63⁻ translucent vesicles in exosome fractions (Fig. 1c) from the SEC procedure, to confirm that A-Exo. indeed stimulates neurite growth, exosomes were depleted from SEC-eluted exosome fractions by CD81 IP or by an additional UC step (100,000 × g for 24 h). Both CD81 IP and the additional UC step effectively depleted exosomes, indicated by the detection of CD81 expression only in CD81 IP and UC pellets but not in flow-through (FT) from CD81 IP or in supernatant (SN) from the UC step (Supplementary Fig. 2e). Consistently, exosome-depleted FT from CD81 IP or SN from the additional UC step has no effect on stimulating neurite growth (Fig. 1d iv, v, e), while the pelleted A-Exo. from the additional UC step retains the stimulatory effect on neurite growth (Fig. 1d iii, e).

Subsequent immunostaining of neurite markers indicates that axons (Map2⁻βIII-tubulin⁺) but not dendrites (Map2⁺βIII-tubulin⁺) are specifically elongated by A-Exo. treatment (Fig. 1f, g and Supplementary Fig. 2f). Active axonal elongation of cortical neurons induced by A-Exo. was also observed in time-lapse live cell imaging (8 h time frame, Supplementary Movie 2 compared to Supplementary movie 1). Immunostaining of additional axon markers such as Tau was also performed to confirm axon-specific stimulation by A-Exo. (Supplementary Fig. 2g). βIII-tubulin staining was then primarily shown for neurite labeling in subsequent results. Interestingly, A-Exo. treatment induces no significant changes in neuronal morphology and synapse numbers (Fig. 1h), indicated by similar neurite VGluT1 and PSD95 density (quantified from secondary branches, Fig. 1i, j). Consistently, Sholl analysis of cortical neurons also confirmed that the overall morphological complexity of cortical neurons is not altered by A-Exo. treatment, other than continuous intersections at distal but not proximal (<150 μm) distances from the soma (Supplementary Fig. 2h), as a result of elongated axons.

### Surface expression of HepaCAM (GlialCAM) mediates stimulatory effect of A-Exo. on axon growth

To begin dissecting how A-Exo. stimulate axon growth, we performed different biochemical treatments, i.e., proteinase K, RNase, and/or sonication on A-Exo., to examine whether RNA or proteins especially surface proteins, mediate the stimulatory effect of A-Exo. on axon growth. To test whether RNA (including microRNA) in exosomes is involved in exosome-mediated stimulation of axon growth, sonicated

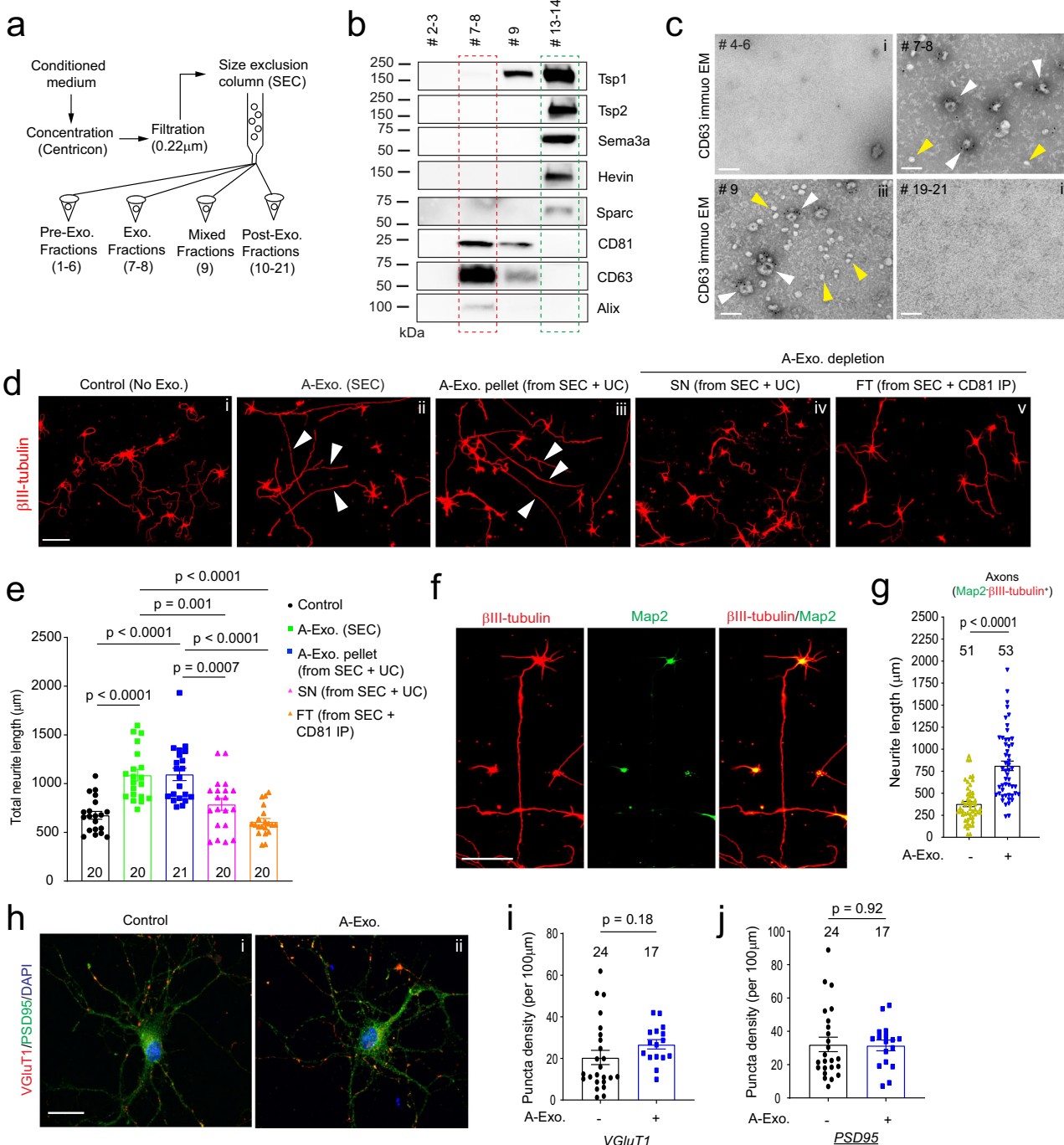

**Fig. 1 | Size exclusion chromatography (SEC)-isolated astroglial exosomes (A-Exo.) selectively stimulate neuronal axon growth. a** Schematic diagram of SEC-based isolation of exosomes from ACM. A 10k molecular weight cutoff Centricon® Plus-70 centrifugal filter device was used; **b** Representative immunoblots (from >5 replicates) of astroglia secreted proteins and exosome markers (CD81, CD63, and Alix) from eluted fractions (pooled as indicated, 500 μl/fraction) of ACM (100 ml/sample) from SEC. Unconcentrated elution (10 μl/sample) was run on immunoblot; **c** Representative (from >3 replicates) immunoEM images of CD63 labeling in different SEC eluted fractions. i–iv: fractions #4–6, #7–8, #9, and #19–21, respectively; white arrows: CD63⁺ A-Exo.; yellow arrows: CD63⁻ small vesicles; scale bar: 100 nm. Representative images (**d**) and quantification (**e**) of βIII-tubulin⁺ neurite length of cortical neurons in control (i) or treated with SEC-isolated A-Exo. (ii) A-Exo. pellet (iii, from SEC + UC), A-Exo. depleted SN (iv, from SEC + UC), or FT (v, from SEC +

CD81 IP); white arrows: elongated neurites; scale bar: 100 μm. Number of neurons quantified in each group shown in the graph (6–8 neurons/replicate, 3 biological replicates)/group; **f** Representative image of βIII-tubulin and Map2 stained cortical neurons following A-Exo. treatment. Scale bar: 100 μm; **g** Quantification of Map2⁻βIII-tubulin⁺ axon length following A-Exo. treatment. *n* = 51 (control), 53 (A-Exo.) neurons (10–13 neurons per replicate, >3 biological replicates)/group; Representative image of VGluT1 and PSD95 staining on cortical neuronal cultures (**h**) and quantification of VGluT1 density (**i**) and PSD95 density (**j**), Scale bar: 30 μm; *n* = 5 neurites/neuron, number of neurons quantified in each group shown in the graph (5–8 neurons/replicate, 3 biological replicates)/group; *p* values in (**e**) determined by one-way ANOVA followed by post hoc Tukey's test; *p* values in (**g, i, j**) determined by two-tailed *t*-test. Data are presented as mean values ± SEM.

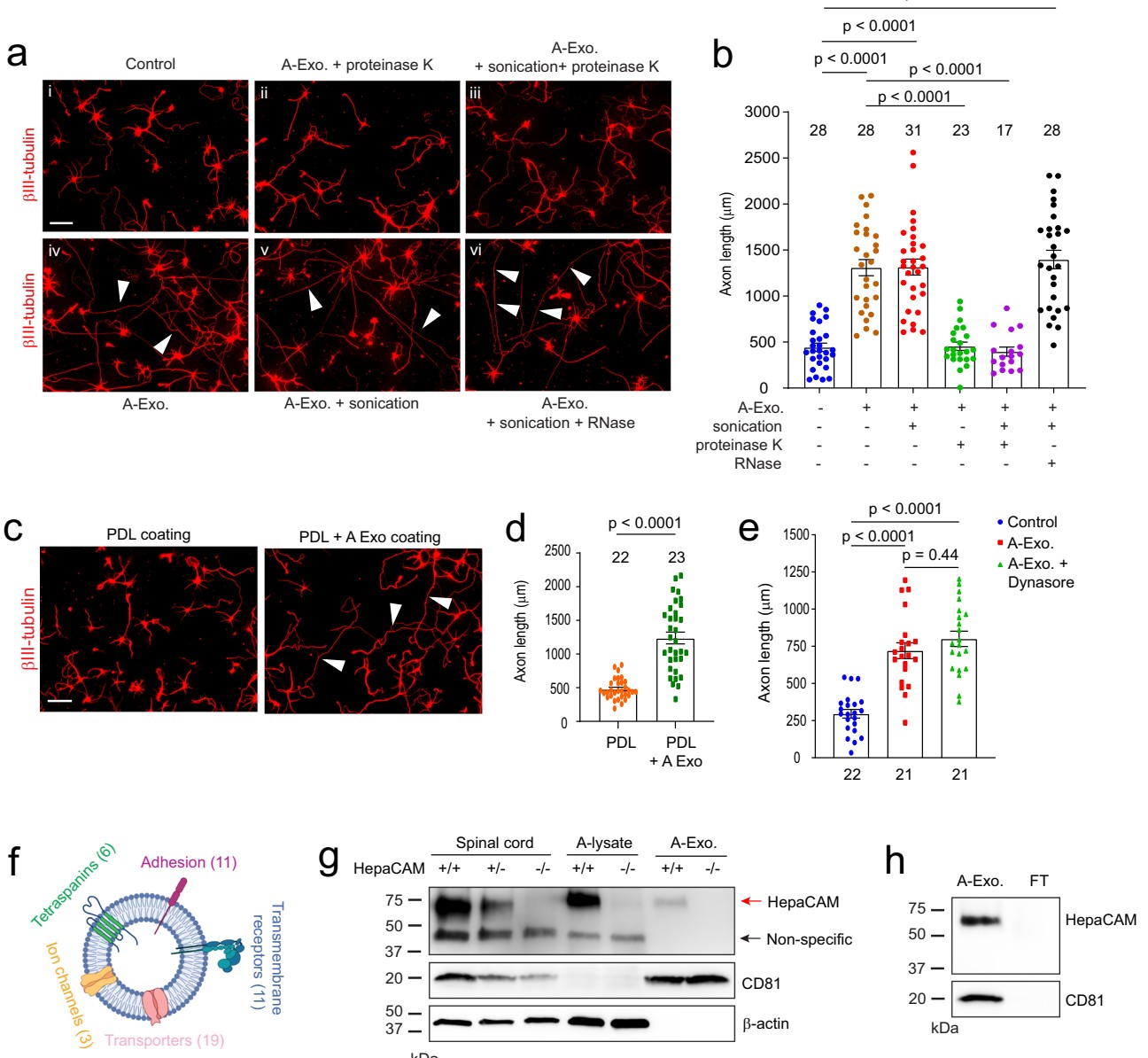

**Fig. 2 | Involvement of A-Exo. surface signals in promoting axon growth and identification of the surface expression of HepaCAM (GlialCAM) on A-Exo.**
Representative images (**a**) and quantification (**b**) of axon length of cortical neurons in control (i) or treated with proteinase K (10 μg/ml, 5 min) digested A-Exo. (ii) sonicated (30 s) and proteinase K digested A-Exo. (iii) A-Exo. (iv) sonicated A-Exo. (v) or sonicated (30 s) and RNase (10 μg/ml, 5 min) digested A-Exo. (vi) 1 μg A-Exo./sample was used in each treatment in (**a**, **b**). White arrows: elongated axons; Number of neurons quantified in each group shown in the graph (6–11 neurons/replicate, 3 biological replicates)/group; Scale bar: 100 μm; Representative images (**c**) and quantification (**d**) of axon length of cortical neurons plated on either poly-D-lysine (PDL) coated or PDL/A-Exo. coated coverslips. Number of neurons quantified in each group shown in the graph (10–11 neurons/replicate, 3 biological replicates)/group; Scale bar: 100 μm; **e** Quantification of axon length of cortical neurons following A-Exo. treatment or co-treatment with A-Exo. and dynasore (dynamin

inhibitor, 50 μM). *n* = 22 (control) or 21 (A-Exo. and A-Exo. + dynasore) neurons (6–7 neurons/replicate, 3 biological replicates)/group; **f** Proteomic identification of different categories of transmembrane proteins on A-Exo. surface. Specific transmembrane proteins are included in Supplementary Table 1. *n* = 3 biological replicates; **g** Detection of specific HepaCAM immunoreactivity (>5 replicates) from spinal cord lysate (10 μg/lane), astrocyte lysate (10 μg/lane), and A-Exo. (1 μg/lane) prepared from WT (+/+), HepaCAM heterozygous (+/−), and HepaCAM KO (−/−) mice; Red arrow: specific HepaCAM immunoreactivity; Black arrow: non-specific immunoreactivity; **h** Detection of specific HepaCAM immunoreactivity (>5 replicates) in A-Exo. but not in exosome-free ACM fractions; 1 μg A-Exo. was used in each experiment. *p* value in (**d**) determined from two-tailed *t*-test; *p* values in (**b**, **e**) determined by one-way ANOVA followed by post hoc Tukey's test. Data are presented as mean values ± SEM.

and RNase treated A-Exo. (1 μg/sample) were added onto cortical neuronal cultures. Interestingly, exosomes with essentially all RNA degraded, as confirmed by bioanalyzer analysis (Supplementary Fig. 3a), are still able to strongly stimulate neurite growth, similarly to untreated A-Exo. (White arrows, Fig. 2a vi, b), supporting the non-involvement of RNA in mediating the stimulatory effect of A-Exo. on axon growth. In contrast, proteinase K treatment of A-Exo. in which surface exosomal

proteins, such as CD81, are degraded (Supplementary Fig. 3b) completely abolished the stimulatory effect of A-Exo. on axon growth (Fig. 2a ii, iii, b). In addition, sonicated A-Exo. surface fractions without lysate remain equally as stimulatory as untreated A-Exo. (White arrows, Fig. 2a iv, v, b). These results point to a potential surface protein mechanism in mediating the stimulatory effect of A-Exo. on axon growth. We further tested the involvement of A-Exo. surface contact

with neurons by plating cortical neurons onto coverslips that were coated with poly-D-lysine (PDL), PDL/laminin (LN), PDL + A-Exo., or PDL/LN + A-Exo. Consistent with the results from biochemical treatments of A-Exo., neuronal axons are significantly longer on PDL or PDL/LN with A-Exo.-coated coverslips compared to PDL or PDL/LN coated alone (Fig. 2c, d and Supplementary Fig. 3c). Additionally, inhibition of clathrin-dependent endocytosis by dynasore, a cell-permeable inhibitor of dynamin[32], has no effect on A-Exo.-stimulated neuronal axon growth (Fig. 2e), excluding the possibility of clathrin-mediated endocytosis of A-Exo. in promoting neuronal axon growth. Together, these results support the notion that surface protein-mediated contact mechanisms mediate the axon-stimulating effect of A-Exo.

Surface proteins (and internal protein cargos) of A-Exo. remain essentially unknown. As molecular cargoes in exosomes are highly heterogeneous and cell-type dependent[20], we performed proteomic analysis on A-Exo. by in-gel trypsin digestion and LC/MS/MS analysis. A total of 347 proteins were identified based on 3 peptides detected per protein and iBAQ $>1 \times 10^5$. We used Ingenuity Pathway Analysis (IPA) to specifically analyze transmembrane proteins detected on A-Exo. and found tetraspanins (exosome markers), cell-adhesion molecules (CAMs), transmembrane receptors, transporters, and channels (Fig. 2f and Supplementary Table 1). In particular, HepaCAM (also named GlialCAM), a transmembrane CAM protein highly enriched in CNS astroglia[16], was found on the surface of A-Exo. Specific HepaCAM immunoreactivity (~70 KDa size but not the non-specific 45 KDa size) was also determined and verified in spinal cord, astrocyte lysate, and A-Exo. samples from WT (+/+), HepaCAM heterozygous (+/−), and KO (−/−) mice (generated from HepaCAM floxed mice[19]) (Fig. 2g). Additionally, HepaCAM was detected only in A-Exo. but not in non-exosome FT in ACM (Fig. 2h), consistent with its characterization as a transmembrane protein. Although the naïve form of HepaCAM protein is predicted to be ~50 KDa, its glycosylated and membrane associated form has been detected at ~70 KDa as shown here and previously[16]. Thus, the detection of the glycosylated but not the naïve form of HepaCAM in exosome samples (Fig. 2g, h) also supports the functional role of HepaCAM on exosomal surface. In addition, although certain transmembrane proteins undergo proteolytic cleavage to release their extracellular domain (ECD)[33], we found no specific HepaCAM immunoreactivity band (~40 KDa) that would correspond with the size of cleaved ECD in our HepaCAM immunoblots (Supplementary Fig. 3d and Fig. 2g, h), ruling out the possibility that HepaCAM undergoes proteolytic cleavage to release its ECD in vitro and in vivo.

Although HepaCAM belongs to the CAM family with Ig-like extracellular domains[15], its involvement in axon growth remains unexplored. We tested whether HepaCAM is involved in mediating the stimulatory effect of A-Exo. on axon growth by treating cortical neurons with HepaCAM-depleted A-Exo., prepared from HepaCAM KO mouse astrocyte cultures. As shown in Fig. 3a, equal amount (1 μg) of HepaCAM-depleted A-Exo. only modestly stimulate neurite growth compared to WT A-Exo. (a 45% reduction, $p < 0.0001$, Fig. 3b), demonstrating the important role of HepaCAM in mediating the axon-stimulating effect of A-Exo. This is consistent with the observation that HEK cell exosomes, which do not stimulate axon growth (Supplementary Fig. 2d), lack HepaCAM expression (Supplementary Fig. 3e). Previous studies have shown that HepaCAM depletion dysregulates proper targeting of surface proteins such as Mlc1 and the chloride channel Clc-2 in glial cells[34]. To further demonstrate that the axon-stimulating effect of A-Exo. is mediated directly by HepaCAM but is not due to mistargeted surface proteins resulting from the HepaCAM depletion, we treated cortical neurons with both HepaCAM antibody and A-Exo. The addition of HepaCAM antibody effectively and completely blocked A-Exo's stimulatory effect on axon growth (Fig. 3c iv, d), while the control IgG antibody had no effect (Fig. 3c iii) on A-Exo's stimulation of axon growth. IgG itself also has no effect on neuronal

axon growth (Fig. 3c ii). To further demonstrate that HepaCAM is sufficient to stimulate axon growth, coverslips were directly coated with PDL and either the HepaCAM extracellular domain (ECD) or BSA. HepaCAM ECD, but not BSA, sufficiently and significantly stimulates axon growth (Fig. 3e iii, f). Together, these genetic and biochemical analyses clearly support the direct role of HepaCAM ECD in mediating the axon-stimulating effect of A-Exo.

## Developmental dynamics and in situ distribution of astroglial exosomes in the CNS

Although a number of in vitro studies have reported secretion of exosomes from cultured astroglia, in situ distribution and developmental changes of A-Exo. in the CNS remain unexplored, primarily due to the difficulty of selectively labeling cell-type specific exosomes by immunostaining. We previously generated cell-type specific exosome reporter hCD63-GFP$^{f/f}$ mice[35], which allows labeling of cell-type specific exosomes and their intracellular precursors, intraluminal vesicles (ILVs) and multiple vesicular bodies (MVBs). We first confirmed secretion of GFP-tagged A-Exo. by immunoblotting GFP and exosome markers Alix and CD81 (Supplementary Fig. 4a) on A-Exo. collected from 4-OHT-treated *Slc1a3*-CreER$^+$hCD63-GFP$^{f/+}$ astrocyte cultures. By employing this mouse tool and confocal/immunoEM imaging, we have previously characterized neuronal ILVs and exosomes in situ in the CNS[35]. To determine the in vivo distribution of A-Exo. in the developing CNS, we generated hCD63-GFP$^{f/+}$Ai14-tdT$^{f/+}$ mice and performed stereotaxic injections of AAV5-mCherry-*Gfap*-Cre (0.3 μl, $4 \times 10^{12}$ gc/ml) on the motor cortex at either P1 or P21 for tissue collection at P6-8 or P28, respectively (Fig. 4a). This combined AAV5-mCherry-*Gfap*-Cre and hCD63-GFP$^{f/+}$Ai14-tdT$^{f/+}$ mice paradigm allows selective labeling of both astroglial morphology and astroglia secreted exosomes (as well as ILVs/MVBs) simultaneously, which facilitates identification of secreted astroglial exosomes from the same labeled astroglia. hCD63-GFP$^+$ puncta were found to be clearly co-localized with tdT$^+$ astroglial soma and processes at both P6-8 and P28 (Fig. 4b and Supplementary Fig. 4b, the orthogonal view of Fig. 4b iii in Supplementary Fig. 4c). By converting confocal images (Fig. 4b i, iii) into 3D images (Fig. 4b ii, iv) using Imaris image analysis software and quantifying extracellular (secreted exosomes, yellow arrows, Fig. 4b i−iv) and intracellular (ILVs/MVBs, white arrows, Fig. 4b iii) hCD63-GFP$^+$ puncta based on tdT$^+$ astroglial labeling, significantly more ($p = 0.0006$) hCD63-GFP$^+$ puncta were observed outside of tdT$^+$ cortical astroglia at P8 (75.5%) when astroglial processes are largely undeveloped[13,36] than P28 (23.0%) when astroglial processes are fully developed (Fig. 4c), suggesting that astroglial exosomes are particularly and abundantly secreted during first postnatal week when astroglial processes are still primitive.

To further examine astroglial exosome distribution and spreading in spinal cord, we performed stereotaxic injections of AAV5-mCherry-*Gfap*-Cre virus (0.5 μl, $4 \times 10^{12}$ gc/ml) into the gray matter of lumbar spinal cord of adult (P90) hCD63-GFP$^{f/+}$ mice. We decided to perform injections on adult mice to better target spinal cord gray matter which is nearly unfeasible in P1 pups. However, we also observed widespread hCD63-GFP$^+$ puncta from astroglia (Supplementary Fig. 4d i) that surround βIII-tubulin$^+$ axons (Supplementary Fig. 4d ii) in longitudinal spinal cord sections of young (P7) *Slc1a3*-CreER$^+$hCD63-GFP$^{f/+}$ mice following a 4-OHT injection (at P2), suggesting that abundant astroglial exosomes are secreted in the spinal cord during the first postnatal week. We also isolated exosome fractions from P3-5 mice brains and spinal cords and detected selective expression of HepaCAM in the exosome fraction. Note that HepaCAM is detected with a C-terminal recognizing antibody as the N-terminal is likely to be damaged by the enzymatic treatment during in vivo exosome isolation. In addition, we performed HepaCAM immunostaining on spinal cord sections of P7 4-OHT-injected *Slc1a3*-CreER$^+$hCD63-GFP$^{f/+}$ mice and observed HepaCAM immunoreactivity co-localized with hCD63-GFP$^+$ puncta (white arrows, Supplementary Fig. 4e i) with no HepaCAM immunoreactivity

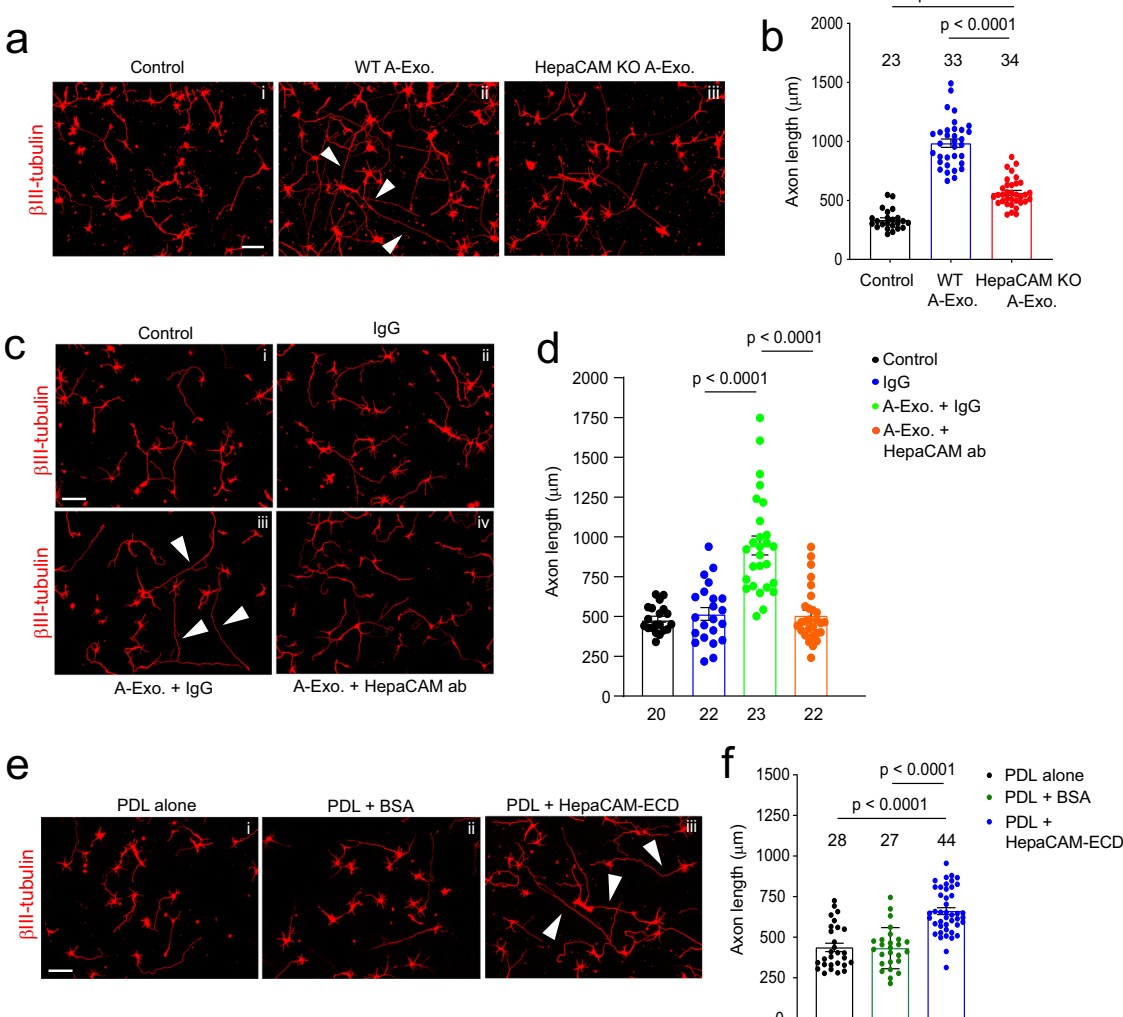

**Fig. 3 | Surface expression of HepaCAM essentially and sufficiently mediates stimulatory effects of A-Exo. on axon growth.** Representative images (**a**) and quantification (**b**) of βIII-tubulin⁺ neuronal axon (white arrows) length following equal amount (1 μg) of WT and HepaCAM-depleted A-Exo. treatment. i control; ii WT A-Exo.; iii HepaCAM KO A-Exo.; HepaCAM-depleted A-Exo. were prepared from HepaCAM KO astrocyte cultures as described in materials and methods. Scale bar: 100 μm; Number of neurons quantified in each group shown in the graph (5–8 neurons/replicate, >3 biological replicates)/group; Representative images (**c**) and quantification (**d**) of βIII-tubulin⁺ neuronal axon (white arrows) length following co-treatment with HepaCAM antibody (ProteinTech) and A-Exo. i control (1 x PBS); ii IgG alone; iii A-Exo. + IgG; iv A-Exo. + HepaCAM ab; 8 μg ab/coverslip (12 mm diameter) was used in the treatment. Scale bar: 100 μm; Number of neurons quantified in each group shown in the graph (7–9 neurons/replicate, ≥2 biological replicates)/group; Representative images (**e**) and quantification (**f**) of βIII-tubulin⁺ neuronal axon (white arrows) length following HepaCAM ECD coating. i PDL alone; ii PDL + BSA (4 μg); iii PDL + HepaCAM ECD (4 μg); Scale bar: 100 μm; Number of neurons quantified in each group shown in the graph (8–12 neurons/replicate, ≥3 biological replicates)/group; 1 μg A-Exo. was used in each experiment; *p* values in (**b**, **d**, **f**) determined using one-way ANOVA followed by a Tukey post hoc test. Data are presented as mean values ± SEM.

detected on HepaCAM KO spinal cord sections (Supplementary Fig. 4e ii). HepaCAM protein expression in spinal cord was observed as early as P0 that also undergoes a similar developmental up-regulation (Supplementary Fig. 4f, g) as in cortex[16].

A single AAV injection into the spinal cord of adult mice results in bright hCD63-GFP⁺ fluorescence at the injection site, indicated by the mCherry fluorescence (yellow arrow, Fig. 4e) expressed from the AAV. By quantifying and calculating the percentage of hCD63-GFP⁺ area (red dashed circle) out of the ventral horn gray matter (GM) area (white dashed circle) on coronal sections, we found that hCD63-GFP⁺ signals spread as far as 4000 μm in each direction along the spinal cord (Fig. 4f) while the AAV (indicated by mCherry) only diffuses around 1000 μm in each direction (Fig. 4e, f). This longitudinal hCD63-GFP⁺ signal analysis from the focal AAV injection suggests that astroglial exosomes are indeed able to spread over long distances. We also performed a single focal injection of AAV-*Gfap*-Cre into lumbar spinal cords of hCD63-GFP^{f/+}Ai14-tdT^{f/+} mice (P60) and similarly observed

intracellular (white arrows) and extracellular (yellow arrows) hCD63-GFP⁺ signals from high-magnified confocal images (Fig. 4g), confirming that GFP⁺ astroglial exosomes can also be secreted in spinal cord astroglia. Induced hCD63-GFP⁺ signals are also found overlapped with specific endogenous mouse (m)CD63 immunoreactivity (white arrows, Supplementary Fig. 4h iv), confirming the faithful labeling of hCD63-GFP on astroglial ILVs/exosomes. Note that mCD63 is not expected to completely overlap with hCD63-GFP, as mCD63 also labels ILVs/exosomes in other CNS cell types. To overcome the detection limit of confocal microscopy, we further examined induced hCD63 signals by immuno-EM in spinal cord sections of AAV5-mCherry-*Gfap*-Cre-injected hCD63-GFP^{f/+} mice. Clustered hCD63⁺ immunogold signals were found not only inside astroglia (yellow arrows, labeled ILVs or MVBs, Fig. 4h ii) but also in post-synaptic (indicated by black arrows) dendritic ("D") terminals (yellow arrows, Fig. 4h iv), further supporting the notion that hCD63-GFP⁺ A-Exo. are indeed able to be secreted extracellularly and subsequently be internalized into neurons.

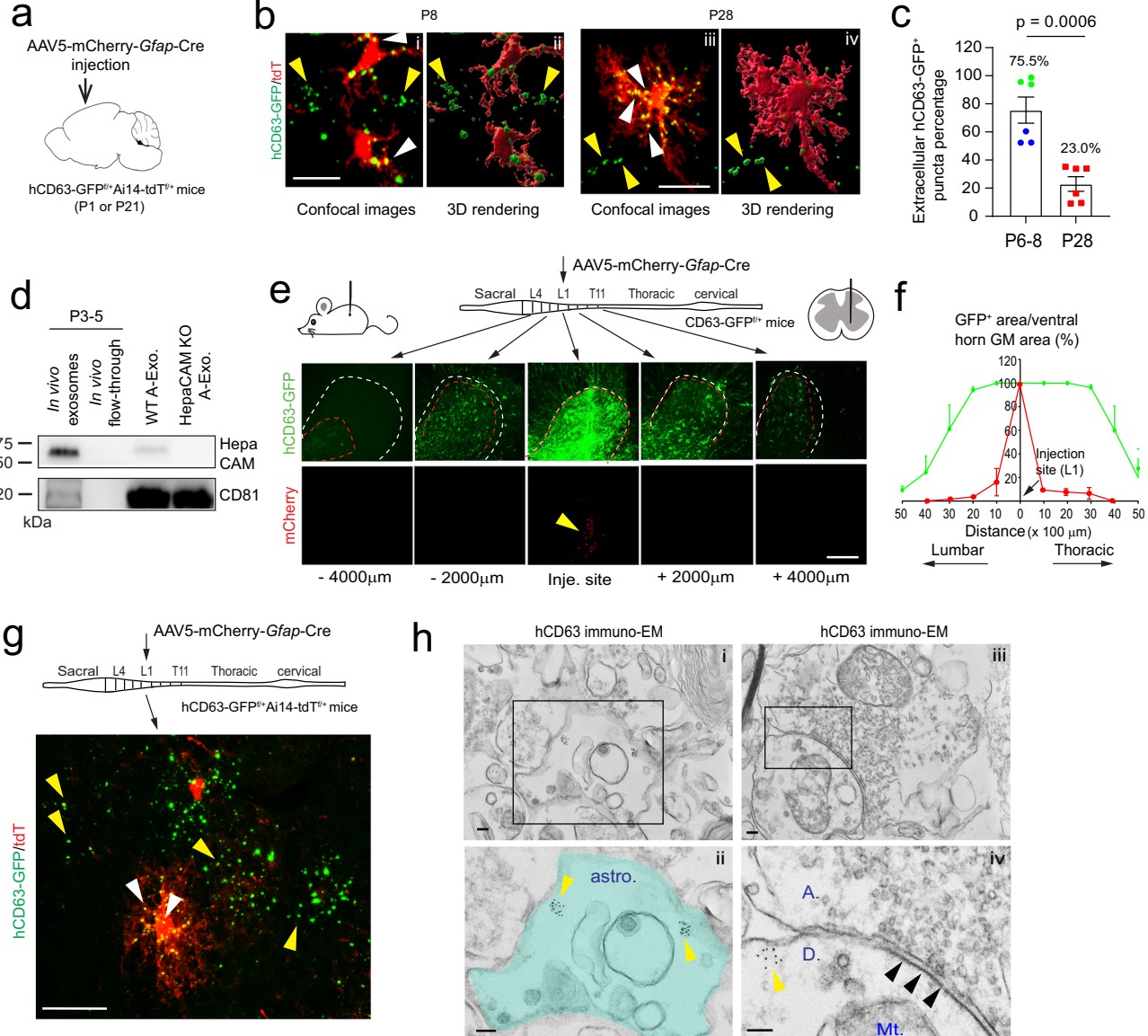

**Fig. 4 | In situ illustration and developmental dynamics of A-Exo. in the CNS.**
**a** Schematic diagram of stereotaxic injections of AAV5-mCherry-*Gfap*-Cre into the motor cortex of hCD63-GFP[f/f]Ai14-tdT[f/f] mice at P1 or P21. Mice were collected for analysis at P6-8 or P28, respectively. **b** Representative confocal and Imaris images of tdT[+] astroglia and hCD63-GFP[+] puncta at P6 (green)–8 (blue) and P28 in AAV5-mCherry-*Gfap*-Cre-injected hCD63-GFP[f/f]Ai14-tdT[f/f] mice. Yellow and white arrows indicate extracellularly or intracellularly localized hCD63-GFP[+] puncta, respectively, based on their co-localization with tdT[+] astroglia; Scale bar: 15 μm (i); 20 μm (iii); **c** Quantification of extracellularly localized hCD63-GFP[+] puncta based on their co-localization with tdT[+] astroglia; n = 18 images (3 images/mouse) from 6 mice/group; **d** Representative (from 3 replicates) HepaCAM immunoblot from in vivo isolated exosome fractions following the SEC purification. P3-5 (3 pups pooled as one sample) pups CNS tissues (brain and spinal cord) were used in exosome isolation. HepaCAM C-terminal antibody was used in detecting HepaCAM; A-Exo. from cultured WT or HepaCAM KO astrocytes were used as positive and negative controls; **e** Schematic view of AAV5-mCherry-*Gfap*-Cre virus injection into spinal cord of hCD63-GFP[f/f] mice (P90) and representative images of induced hCD63-GFP[+] and mCherry signals in proximal and distal spinal cord sections from the injection site. Mice analyzed 2 weeks post-injection; Red dashed circles: hCD63-GFP[+] area; White dashed circles: ventral horn gray matter area; mCherry signals are only visible within 500 μm from the injection site. Scale bar: 200 μm; **f** Quantification of the distance hCD63-GFP[+] and mCherry signal traveled along spinal cord in AAV5-mCherry-*Gfap*-Cre injected hCD63-GFP[f/f] mice. n = 4 mice (injected at P90); **g** Schematic view of AAV5-mCherry-*Gfap*-Cre virus injection into spinal cords of hCD63-GFP[f/f]Ai14-tdT[f/f] mice (P60) and the representative (from >3 injected mice) magnified image of induced hCD63-GFP[+] and tdT[+] astroglia; Yellow and white arrows indicate extracellularly or intracellularly localized hCD63-GFP[+] puncta, respectively; Scale bar: 20 μm; **h** Representative (from 3 mice) immunoEM images of hCD63 labeling on spinal cord sections of AAV5-mCherry-*Gfap*-Cre-injected hCD63-GFP[f/f] mice. Intracellular immunogold signals (yellow arrows) are observed inside astroglia (astro., i) and in neuronal post-synaptic (indicated with black arrows, iv) dendritic compartment (D, iv). ii and iv are the magnified views of i and iii, respectively. A axonal terminal, Mt mitochondria; Scale bars, 100 nm. *p* value in (**c**) determined from two-tailed *t*-test. Data are presented as mean values ± SEM.

## HepaCAM is important for early postnatal CST axon growth and promotes growth cone size

Although developing axon growth in the CNS is mostly completed at birth in mice, CST axons continue to grow especially during the 1st postnatal week[2] (Fig. 5a diagram) during which A-Exo. are abundantly secreted (Fig. 4b). Anterograde tracing dyes, such as CM-DiI, have been previously used[37] to label layer V pyramidal neurons in the motor cortex and their descending axons, especially during early postnatal development (representative labeling image in Fig. 5b), which allows tracing of their continuous postnatal growth. Other genetic approaches, such as Emx1-Cre × Thy1-STOP-YFP[38] or UCHL1-eGFP[39] mice, are specifically suitable for adult but not developing CST labeling and can

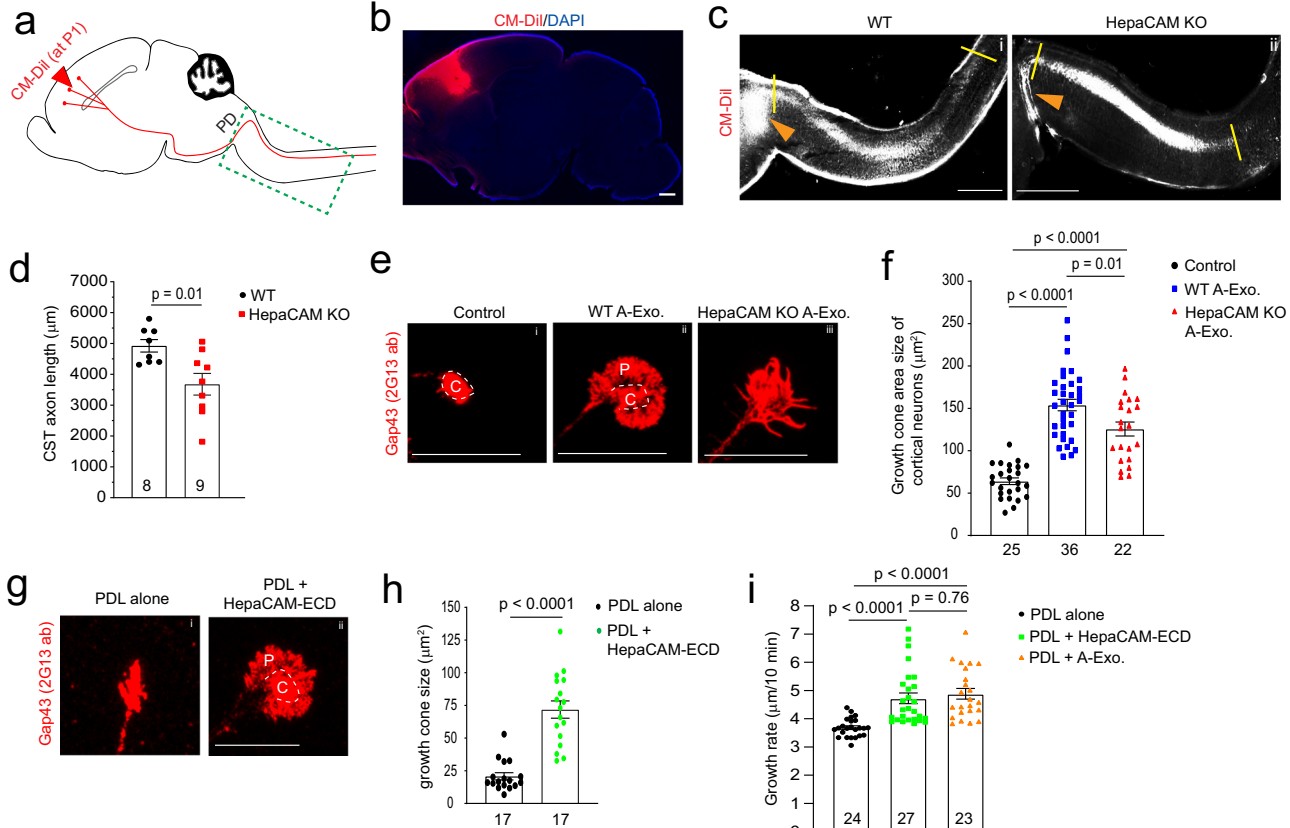

**Fig. 5 | HepaCAM is essential for early postnatal CST axon growth and expands axon growth cone size. a** Diagram of CM-DiI dye injections at the motor cortex to label layer V pyramidal neurons and descending CST axons; PD pyramidal decussation; green dashed box indicates postnatal CST growth (as shown in **c**); **b** Representative (from >10 injected mice) image to show the CM-DiI labeling in the motor cortex 2 days following the injection; Scale bar: 1 mm; Representative images (**c**) and quantification (**d**) of CM-DiI-labeled CST axons in the spinal cord of WT (i) and HepaCAM KO (ii) mice. Orange arrows indicate the pyramidal decussation; yellow lines indicate the beginning and ending points for the CST axon length measurement; The image was generated by superimposing images of serial longitudinal sections, which are shown in Supplementary Fig. 5a. Scale bar: 1 mm; $n = 8$ mice for WT and 9 mice for HepaCAM KO; Representative images (**e**) and quantification (**f**) of axon growth cone size of control (i) cortical neurons or cortical neurons treated with WT (ii) or HepaCAM KO (iii) A-Exo. C center domain (white circle), P peripheral domain (growth cone area outside of the center domain); Scale bars: 20 μm. Number of neurons quantified in each group shown in the graph (4–14 neurons/replicate, 3 biological replicates)/group; Representative images (**g**) and quantification of axon growth cone size (**h**) of cortical neurons grown on either PDL alone (i) or PDL/HepaCAM-ECD (ii) coating. $n = 17$ neurons (5–8 neurons/replicate, 3 biological replicates)/group; Scale bar: 20 μm; **i** Quantification of axon growth rate of cortical neurons treated with HepaCAM ECD and A-Exo. Number of neurons quantified in each group shown in the graph (7–9 neurons/replicate, 3 biological replicates)/group; live-cell imaging was performed at DIV 3 for 2 h; $p$ value in (**d**, **h**) determined from two-tailed $t$-test; $p$ values in (**f**, **i**) determined using one-way ANOVA followed by a Tukey post hoc test. Data are presented as mean values ± SEM.

also be non-specific[40]. We therefore performed focal CM-DiI dye injections on the layer V motor cortex of WT and HepaCAM KO pups (P1). Pups were collected 48 h post injection and longitudinal sections of the spinal cord were prepared as shown in Supplementary Fig. 5a. This time point was chosen to facilitate the preparation of longitudinal spinal cord sections and to observe consistent DiI labeling. CM-DiI-labeled CST axons undergo pyramidal decussation (PD, orange arrows, Fig. 5c) and continue to elongate in spinal cords. The representative images (Fig. 5c) were created by superimposing multiple individual images taken from longitudinal spinal cord sections from lateral to middle orientation (Supplementary Fig. 5a, b). CST axons cross the midline and begin to enter the spinal cord at birth in mice. We therefore quantified the length (between two yellow lines, Fig. 5c i, ii) of CM-DiI-labeled CST axons that grow into the spinal cord from the PD. Quantitative measurement found that CST axons grow a significantly (~1300 μm, $p = 0.01$) shorter distance into the spinal cord of HepaCAM KO pups when compared to WT mice from P1 to P3 (Fig. 5d). This is consistent with in vitro results that HepaCAM-deficient A-Exo. only modestly promotes axon growth compared to HepaCAM-expressing A-Exo. (Fig. 3a). In parallel, a recent study showed that the loss of HepaCAM in astroglia has no effect on density of excitatory

intracortical or thalamocortical synapses and only modestly decreases the density of inhibitory synapses in layer I cortex[19].

Axon elongation is primarily driven by the growth cone, which is composed of central and peripheral domains[9]. In particular, the peripheral domain of the axon growth cone is abundant with actin filament-organized filopodia and lamellipodia[9]. It has been well established that actively elongating axons, such as under nerve growth factor (NGF) stimulation, have increased peripheral domain size in growth cones, an indication of extended filopodia and lamellipodia[41], that increases their contact with surrounding substrates[9]. In contrast, collapsed growth cones have these filopodia and lamellipodia retracted, leading to reduced or even lost peripheral domains[42]. To determine whether A-Exo. alter axon growth cone morphology and especially peripheral domain size, we performed immunostaining of growth associated protein 43 (GAP43, 2G13 clone antibody) that specifically labels axon growth cones[43] following A-Exo. treatment of cortical neurons. Co-immunostaining of Tau and Map2 confirmed specific axon growth cone labeling revealed by the 2G13 antibody (Supplementary Fig. 5c). As expected, axon growth cones from untreated control neurons (DIV 6) have minimal peripheral domains (Fig. 5e i) following initial growth on PDL/laminin (LN) coated

coverslips. A-Exo. treatment induces an enlarged fan-shaped growth cone morphology with extended peripheral domain ("P", Fig. 5e ii), which is a characteristic growth cone morphology induced by CAM substrates, but not LN substrates, which leads to multiple and long protrusions of filopodia in the peripheral domain[41,44]. In contrast, growth cone morphology of neurons treated with HepaCAM-deficient A-Exo. tends to have protrusions of filopodia (Fig. 5e iii) and the total growth cone size is also significantly reduced (Fig. 5f). To directly test the effect of HepaCAM on axon growth cone morphology and size, we next examined axon growth cones of cortical neurons cultured on PDL alone (to minimize the influence of the laminin substrate on growth cone) or on PDL/HepaCAM ECD coating. Consistent with the strong stimulation of axon growth by HepaCAM ECD coating (Fig. 3e, f), HepaCAM ECD induces the formation of a large peripheral domain in growth cones (Fig. 5g ii). The overall growth cone size of neurons treated with HepaCAM ECD is 3-fold larger ($p < 0.0001$) than that of neurons grown on PDL alone (Fig. 5h). To confirm that HepaCAM ECD-induced growth cone size and morphology change is functional, we further set up live time-lapse imaging from neuronal cultures treated with HepaCAM ECD and A-Exo. and found that HepaCAM ECD stimulates axon growth in a comparable rate as by A-Exo. (Fig. 5i). Collectively, these results on changes in axonal growth cone morphology and size support the direct function of HepaCAM in regulating axonal growth cone and promoting axon growth.

## ApoE in non-exosome ACM fractions inhibits A-Exo.-mediated stimulation on neuronal axon growth

Although HepaCAM on A-Exo. robustly stimulates axon growth, synaptogenesis but not axon growth was primarily observed in neurons stimulated by ACM or co-cultured with astrocytes in previous studies[45]. We also confirmed that non-exosome FT from ACM has no stimulatory effect on axon growth (Fig. 1d). Intrigued by these observations, we decided to test the possibility that non-exosome fractions of ACM may suppress A-Exo's effect on axon growth. Interestingly, mixing of 0.2x (concentrated from 2 ml, 70 μg proteins) and 0.5x (concentrated from 5 ml, 175 ug proteins) non-exosome ACM flow-through (FT) from the SEC column with A-Exo. completely abolishes A-Exo's stimulatory effect on axon growth (Fig. 6a, b). Subsequent immunoblotting further found that ApoE and ApoJ, two known apolipoproteins secreted from astrocytes, are either mostly (>98% for ApoE) or completely (ApoJ) detected only in non-exosome fractions of ACM (Fig. 6c) with very low ApoE (but not ApoJ) immunoreactivity detected in A-Exo. only after oversaturated exposure (Supplementary Fig. 6a). Other apolipoproteins, such as ApoB, were not detected in ACM (Fig. 6c). We further isolated in vivo exosomes from P10 brain and spinal cord tissues. Consistently, ApoE was only detected in flow-through samples but not in exosome fractions following the SEC purification (Supplementary Fig. 6b). We then mixed human (h)APOE3 with A-Exo. and found that hAPOE3 is able to dose-dependently abolish the stimulatory effect of cortical (Fig. 6d, e) and spinal cord (Supplementary Fig. 6c) A-Exo. on axon growth. The inhibitory dose of hAPOE3 (starting at 10 μg/ml) is comparable to the ApoE concentration in ACM (~15 μg/ml) based on the densitometry of ApoE immunoblotting with human APOE and ACM samples. Previously, three major APOE protein isoforms, APOE2, 3, and 4, have been identified that are closely associated with Alzheimer's disease (AD) risks in human[46]. However, these differential APOE protein isoforms equally and strongly inhibit A-Exo's stimulatory effect on axon growth (Fig. 6f). In addition, this inhibitory effect is specifically mediated by hAPOE but not by hAPOB or hAPOJ (Supplementary Fig. 6d).

Lipids, including cholesterol, are important structural building blocks for developing axon growth[47]. Although lipids are primarily synthesized within neurons (either in cell bodies or locally at axons) and anterogradely transported to axons for developmental growth[47], during axon regeneration, ApoE, the primary cholesterol carrier, has

been shown to contribute to axon growth[48]. We directly added ApoE, cholesterol, and hHDL separately to cultured neurons to test whether they can stimulate axon growth. Interestingly, none of these treatments had any effect in promoting axon growth of cortical neurons (Supplementary Fig. 6e). Additionally, co-treatment of neurons with A-Exo. and receptor associated protein (RAP), a competitive inhibitor for ApoE binding to its receptor low density lipoprotein receptor-related protein 1 (LRP1) for cholesterol delivery, also has no effect on A-Exo's stimulation of axon growth (Supplementary Fig. 6f, g), suggesting that ApoE/cholesterol does not mediate the stimulatory effect of A-Exo. on axon growth. This is also consistent with the very low level of ApoE detected on A-Exo. (Supplementary Fig. 6a). To confirm that ApoE in ACM indeed inhibits A-Exo's stimulatory effect on axon growth, we collected ApoE-deficient ACM from ApoE KO mouse pups. The loss of ApoE in ApoE KO ACM and A-Exo. was confirmed by immunoblot (Supplementary Fig. 6h). Consistently, FT from the ApoE KO ACM has no inhibitory effect on A-Exo's effect on axon growth while wild type (WT) FT completely inhibits A-Exo's effect on axon growth (Fig. 6g, h). As we showed above that HepaCAM is essential in mediating A-Exo's axon growth stimulation, we then tested whether ApoE physically binds to HepaCAM to block its interaction with neurons. However, no ApoE was detected in HepaCAM pull-down from astroglial cell lysate, while in the IgG control HepaCAM was not pulled down nor ApoE was detected (Supplementary Fig. 6i), which suggests no direct binding between HepaCAM and ApoE. Meanwhile, ApoE KO A-Exo. stimulate axon growth similarly as WT A-Exo. (Fig. 6i, j), further suggesting that ApoE is not involved in mediating A-Exo's stimulatory effect on axon growth. Taken together, these results demonstrate that ApoE is minimally found in A-Exo. and not involved in A-Exo's stimulatory effect on axon growth; rather, ApoE is highly abundant in the non-exosome ACM fraction that strongly inhibits A-Exo's stimulatory effect on axon growth.

## ApoE deficiency reduces developmental synaptogenesis and dendritic spine formation on cortical pyramidal neurons in vitro and in vivo

ApoE-mediated transport of cholesterol to neurons has been shown to promote synaptogenesis in cultured RGCs[4]. To examine whether this ApoE/cholesterol pathway is also essential for cortical neuronal synaptogenesis, WT cortical neurons were treated with ACM collected from WT or ApoE KO astrocyte cultures. The loss of ApoE leads to accumulated cholesterol in cultured ApoE KO astrocytes, indicated by Filipin 3 staining (Supplementary Fig. 7a iii, b), similar to the results of treatment with U18666A, an inhibitor for cholesterol transport, in WT astrocyte cultures (Supplementary Fig. 7a ii), suggesting a reduced secretion of cholesterol from ApoE-deficient astrocytes. Consequently, cortical neurons treated with WT ACM have strongly increased VGluT1 ($p < 0.0001$) but not PSD95 ($p = 0.23$) puncta density on the neurites (Fig. 7a ii, b, c). In comparison, a substantially reduced though still significant ($p = 0.006$) increase in VGluT1 and unchanged PSD95 ($p = 0.24$) puncta density was observed in neurites of cortical neurons treated with ApoE-deficient ACM. Co-localization analysis of VGluT1 and PSD95 along neurites found much reduced synapse density in neurons treated with ApoE KO ACM compared to neurons treated with WT ACM (Supplementary Fig. 7c), suggesting that the astroglial ApoE/cholesterol pathway similarly promotes synaptogenesis of cortical neurons.

ApoE is known to be mostly expressed in and secreted from astroglia in the homeostatic CNS and ApoE mRNA was found to be strongly up-regulated in astroglia during postnatal development by single cell sequencing[49]. We performed ApoE immunoblotting and also found that ApoE protein is only lowly expressed at birth and is robustly up-regulated in cortical tissues during postnatal development (Supplementary Fig. 7d, e). To directly examine whether ApoE deficiency affects dendritic branching and developmental dendritic spine

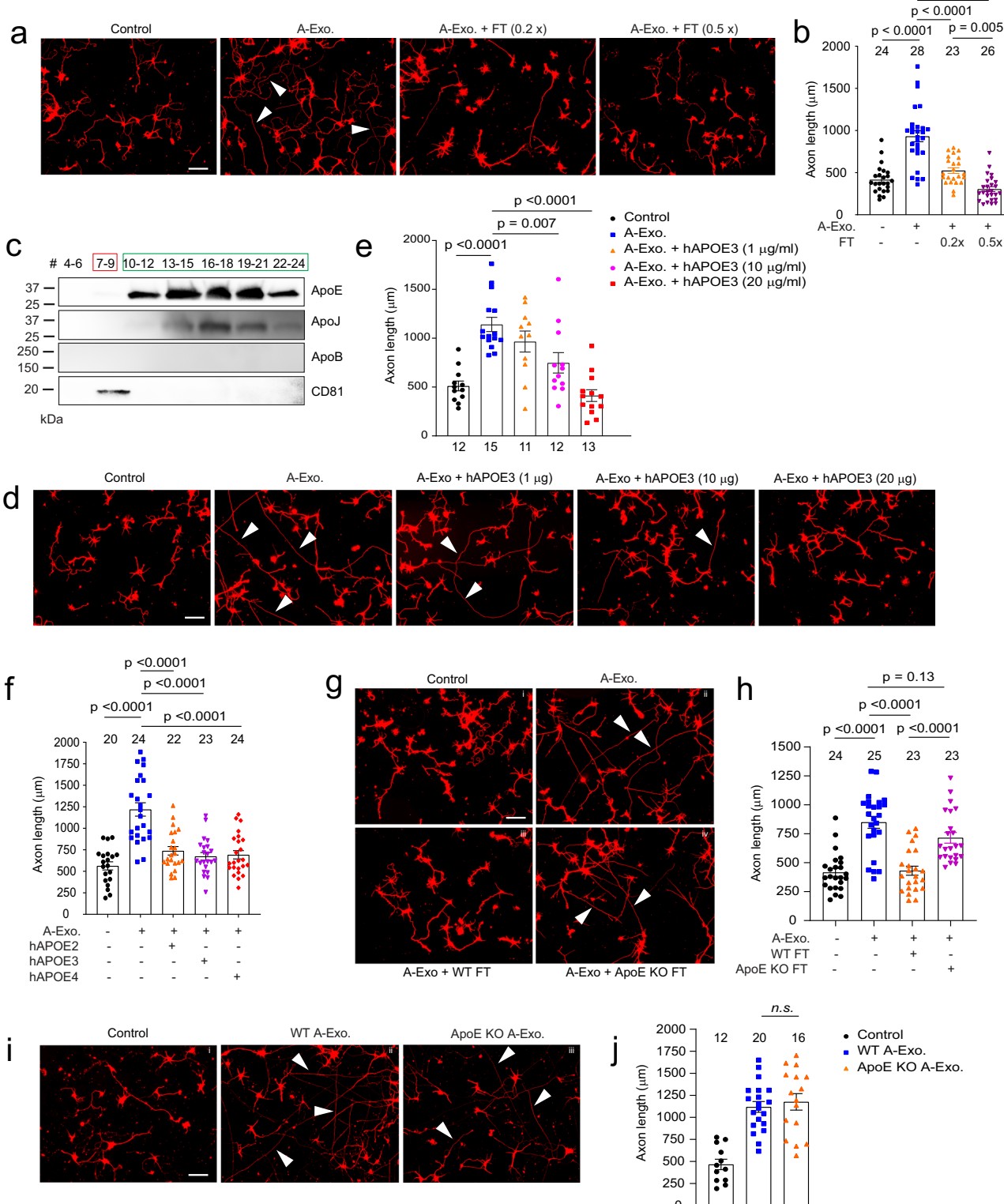

formation of cortical neurons especially layer V pyramidal neurons in the motor cortex, we generated Thy1-YFP⁺ApoE⁻/⁻ mice and analyzed pyramidal neuronal morphology and their dendritic spine density between Thy1-YFP⁺ and Thy1-YFP⁺ApoE⁻/⁻ mice. Thy1-YFP (H line) mice have been widely used to illustrate neuronal morphology including dendritic spines[50]. We observed well-labeled neurons across the CNS including pyramidal neurons in layer V motor cortex (Supplementary Fig. 7f i, ii). The clear YFP labeling also facilitates clear identification of apical and basal dendrites and spines (Fig. 7d, e). By using the filament

tracing function in Imaris software, representative YFP⁺ dendritic spines from individual layer V pyramidal neurons (Supplementary Fig. 7g) in both Thy1-YFP⁺ and Thy1-YFP⁺ApoE⁻/⁻ mice were traced and quantified. Consistent with our in vitro results (Fig. 7a–c), ApoE deficiency leads to substantially reduced spine density on both apical and basal dendrites (Fig. 7f, g) of layer V pyramidal neurons in Thy1-YFP⁺ApoE⁻/⁻ mice. Unexpectedly, the loss of ApoE also increased secondary dendritic branches in Thy1-YFP⁺ApoE⁻/⁻ mice, based on 3D Sholl analysis (Fig. 7h, i), likely compensating for the reduced dendritic spine

**Fig. 6 | ApoE in non-exosome ACM fractions inhibits A-Exo.-mediated stimulation on neuronal axon growth.** Representative images (**a**) and quantification (**b**) of βIII-tubulin⁺ neuronal axon (white arrows) length following treatment of cortical neurons with A-Exo. or A-Exo. mixed with flowthrough (FT) from the SEC column; 0.2x and 0.5x FT each is concentrated from 2- or 5-ml exosome-free ACM, respectively. Number of neurons quantified in each group shown in the graph (7–9 neurons/replicate, 3 biological replicates)/group; Scale bar: 100 μm; **c** Representative (from >5 replicates) immunoblot of different apolipoproteins in all eluted fractions (500 μl/fraction, pooled as indicated) of ACM (100 ml) from SEC with optimal exposure. Unconcentrated elution (15 μl/sample) was run on immunoblot; Representative images (**d**) and quantification (**e**) of βIII-tubulin⁺ neuronal axon (white arrows) length following co-treatment of cortical neurons with A-Exo. and different dose of hAPOE3. Number of neurons quantified in each group shown in the graph (4–5 neurons/replicate, 3 biological replicates)/group; Scale bar: 100 μm; **f** Quantification of βIII-tubulin⁺ neuronal axon length following co-treatment of A-Exo. with common hAPOE isoforms. Number of neurons quantified in each group shown in the graph (10–12 neurons/replicate, 2 biological replicates)/group; Representative images (**g**) and quantification (**h**) of βIII-tubulin⁺ neuronal axon (white arrows) length in control cortical neurons (i) or neurons treated with A-Exo. (ii) and A-Exo. mixed with WT (iii) or ApoE KO (iv) FT, respectively. Scale bar: 100 μm; Number of neurons quantified in each group shown in the graph (7–9 neurons/replicate, 3 biological replicates)/group; Representative images (**i**) and quantification (**j**) of βIII-tubulin⁺ neuronal axon (white arrows) length in control (i) cortical neurons or neurons treated with WT (ii) or ApoE KO (iii) A-Exo. Number of neurons quantified in each group shown in the graph (4–8 neurons/replicate, 3 biological replicates)/group; Scale bar: 100 μm; 1 μg A-Exo. was used in each treatment. *p* values in (**b, e, f, h, j**) were calculated using one-way ANOVA followed by a Tukey post hoc test; n.s. not significant. Data are presented as mean values ± SEM.

density. We further performed CM-DiI injections on ApoE KO pups (P1) to examine whether the loss of ApoE also affects postnatal axon growth, as part of CST extension to the spinal cord, of layer V pyramidal neurons in the motor cortex. We only observed modestly reduced CST axon growth (average ~600 μm shorter, Fig. 7j) but not statistically significant (*p* = 0.36, Fig. 7k) in ApoE KO pups compared to WT pups. This is not unexpected, as ApoE expression is quite low at birth, similar to ApoE KO, and only gradually increased in spinal cords in the first postnatal week during which CST axon growth occurs (Supplementary Fig. 7h, i). In addition, we confirmed that there are no obvious changes of HepaCAM protein expression in ApoE KO mouse cortex, nor no ApoE protein expression changes in HepaCAM KO mice (Supplementary Fig. 7j).

## Discussion

In our current study, by employing an optimized SEC-based exosome isolation procedure, we defined an astroglial exosome-dependent regulatory pathway that stimulates developmental pyramidal neuronal axon growth. This pathway is specifically mediated by astroglial exosomes, as exosome-depleted ACM fractions have no effect in stimulating axon growth. The stimulating effect is axon-specific with a primary action on axon growth cones but not affecting dendritic arborization, length, and synaptogenesis. Consistently, SEC-isolated astroglial exosomes are minimally associated with known astroglia-derived soluble proteins that regulate synaptogenesis. This further supports the notion that astroglial exosomes represent a distinct and unique class of secreted signals from astroglia, in contrast to astroglial secretion of soluble proteins and small molecules to modulate synaptogenesis/maturation and transmission[3]. It is noteworthy to point out that astroglia secreted exosomes can be heterogenous from which HepaCAM-expressing vesicles and other vesicles may have diverse effects on neurons.

Although trophic factors such as NGF/BDNF, are well established to potently promote axon growth[47], our proteomic analysis found no trophic factors in astroglial exosomes, ruling out their involvement in astroglial exosome-stimulated axon growth. Our results also showed that neither RNA mechanisms nor endocytosis are involved in the axon growth-stimulating effect of astroglial exosomes, which is distinct from previous reports that miRNA signals can mediate the axon growth-stimulating effect of mesenchymal stem cell (MSC) exosomes or regulate dendrite complexity through endocytosis[25,51]. Instead, our results provided evidence for an essential and sufficient role of surface HepaCAM on astroglial exosomes in promoting axon growth, representing a unique surface contact mechanism for exosome action. Based on the smaller size of vesicles, HepaCAM⁺ vesicles are likely to be exosomes derived from MVBs, which will be further confirmed in future studies. Nevertheless, this provides a mechanism for plasma membrane surface proteins to be secreted through the MVB pathway, similar to the secretion of transferrin receptors in reticulocytes[52].

These prior studies and our results indicate a growing understanding of the diverse mechanisms and effects of cell-type specific exosomes. Our results also revealed an important function of HepaCAM to mediate intercellular signaling between astroglia and neuronal axons, in addition to its intracellular role as a binding partner to facilitate proper targeting of anion and chloride channels on glial cell surface[17,34] and regulate boundary of neighboring astroglia[19]. How HepaCAM activates downstream pathways in neurons to expand the surface area of growth cones and to promote axon growth remains unclear. CAM protein-mediated downstream signaling is highly diverse and complex by either activating receptors such as integrins, FGF receptors or directly binding intercellularly[53]. As HepaCAM ECD is sufficient to stimulate axon growth, it is possible that HepaCAM ECD activates its neuronal receptor, which remains to be identified, for downstream signaling. In axon growth cones, anterograde polymerization of actin filaments (F-actins) contributes to retrograde flow of F-actin and pushes the growth cone in the forward direction[9]. Previous studies have identified several kinases, particularly focal adhesion kinase (FAK), that are activated downstream of CAM proteins, to promote actin polymerization and axon growth[54,55]. Whether these pathways are involved in HepaCAM ECD's axon-stimulating effect will be investigated in future studies.

Although astroglia are able to secrete various EVs, these previous studies were almost exclusively carried out in cultures[25,26,28]. By employing our previously generated cell-type specific exosome reporter mice and Ai14 reporter mice, our results illustrated the in situ localization and dynamics of secreted A-Exo. in the motor cortex during development and in adult spinal cord. Our results showed that astroglial exosomes are able to spread long distances (up to 8000 μm bidirectionally). In particular, our results showed that A-Exo. can be abundantly localized outside of astroglia during the 1st postnatal week when astroglial morphology remains primitive with limited processes. These extracellularly localized A-Exo. may serve as an alternative cell to cell contact mechanism, especially in the 1st postnatal week, to allow long-range spreading of surface contact signals, such as HepaCAM, via A-Exo. Thus, surface expressed HepaCAM on A-Exo. (Fig. 8) can be a mobile astroglial CAM signal to stimulate CST axon growth postnatally. The detection of HepaCAM expression from in vivo isolated exosomes during early postnatal development further supports this notion. We envision that both exosome and cell surface HepaCAM are involved in stimulating CST axon growth during early postnatal development. As many synapses (both excitatory and inhibitory) are not ensheathed by astroglial processes even in the adult CNS[3,56], mobile surface contact signals on A-Exo. may mediate specific intercellular signaling, in addition to direct plasma membrane contact or the cleavage of transmembrane protein signals.

ApoE is the major carrier for transporting cholesterol and phospholipids in the CNS. It has been extensively studied in CNS pathology,

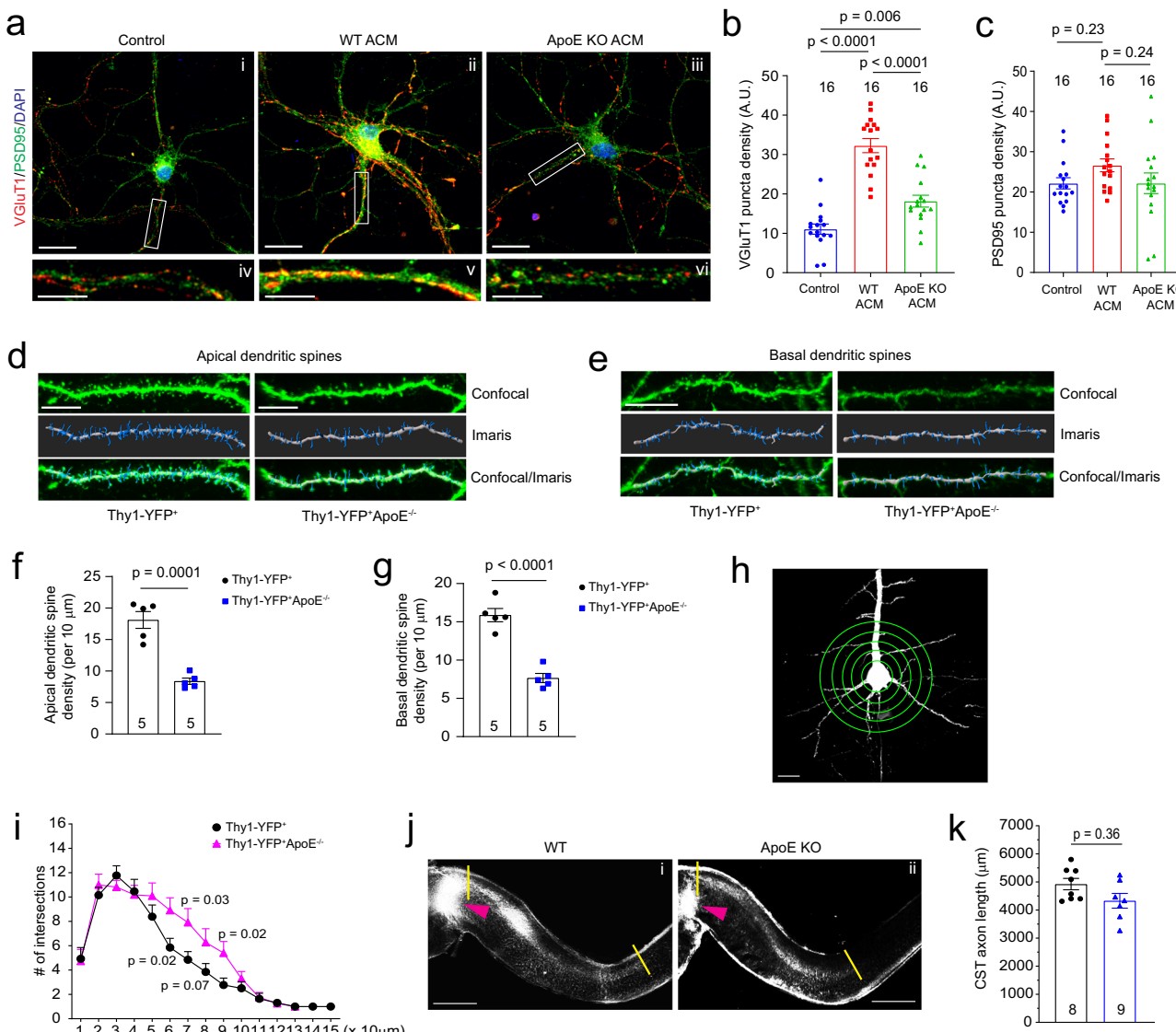

**Fig. 7 | ApoE deficiency reduces developmental dendritic spine formation and alters dendritic branching on layer V cortical pyramidal neurons.** Representative image of VGluT1 and PSD95 staining in cortical neuronal cultures (**a**) and quantification of VGluT1 (**b**) and PSD95 density (**c**) on neurites following ACM treatment. Control cortical neurons (i) and neurite (iv), cortical neurons (ii) and dendrite (v) treated with WT ACM, and cortical neurons (iii) and dendrite (vi) treated with ApoE KO ACM; Scale bar: 20 μm (i–iii) and 10 μm (iv–vi); $n$ = 16 neurons (8 neurons/replicate, 2 biological replicates)/group; Representative confocal and Imaris images of apical (**d**) and basal (**e**) dendrites and spines of layer V pyramidal neurons from motor cortex of Thy1-YFP⁺ and Thy1-YFP⁺ApoE⁻/⁻ mice (P30). Dendrites and spines were traced and quantified in Imaris. Scale bars: 10 μm; Quantification of apical (**f**) and basal (**g**) dendrites of layer V pyramidal neurons from motor cortex of Thy1-YFP⁺ and Thy1-YFP⁺ApoE⁻/⁻ mice (P30). $n$ = 5 mice/group; Representative neuron image (**h**) and 3D Sholl analysis (**i**) of layer V pyramidal neurons from motor cortex of Thy1-YFP⁺ and Thy1-YFP⁺ApoE⁻/⁻ mice. Scale bar: 20 μm; $n$ = 5 mice/group; Representative images (**j**) and quantification (**k**) of CM-DiI-labeled CST axons in the spinal cord of WT (i) and ApoE KO (ii) mice. Orange arrows indicate the pyramidal decussation; yellow lines indicate the beginning and ending points for the CST axon length measurement; Scale bar: 1 mm; $n$ = 8 mice for WT and 9 mice for ApoE KO; $p$ value in (**f**, **g**, **k**) determined by two-tailed $t$-test; $p$ values in (**b**, **c**) determined using the one-way ANOVA followed by a Tukey post hoc test; $p$ values in (**i**) determined using the multiple $t$-test. Data are presented as mean values ± SEM.

and human APOE polymorphism has been closely associated with AD pathogenesis[46]. However, the developmental role of ApoE has not been examined in vivo, despite an early study suggesting that ApoE-mediated transport of cholesterol promotes synaptogenesis in cultures[4]. Our results from both genetic and pharmacological approaches showed that the ApoE/cholesterol pathway is not involved in mediating A-Exo's stimulatory effect on axon growth. On the contrary, abundant ApoE levels are only found in non-exosome ACM fractions that strongly inhibit the stimulatory effect of A-Exo. on axon growth. As ApoE expression is developmentally up-regulated in both spinal cord and cortex, as shown here and previously[49], it may have region-specific and distinct roles in regulating postnatal development

of layer V pyramidal neurons in the motor cortex (Fig. 8). Developmentally increased ApoE in the spinal cord could gradually inhibit A-Exo's stimulatory effect on axon growth, consistent with the observation that ApoE dose-dependently inhibits A-Exo. on axon growth (Fig. 6). On the other hand, developmentally increased ApoE in the cortex promotes cholesterol transport and subsequent dendritic spine formation in the same group of layer V pyramidal neurons in the motor cortex (Fig. 8). Indeed, ApoE deficiency leads to significantly reduced spine density on both apical and basal dendrites of layer V pyramidal neurons in the motor cortex of ApoE KO mice. These results provide important insights about the function of ApoE during postnatal CNS development.

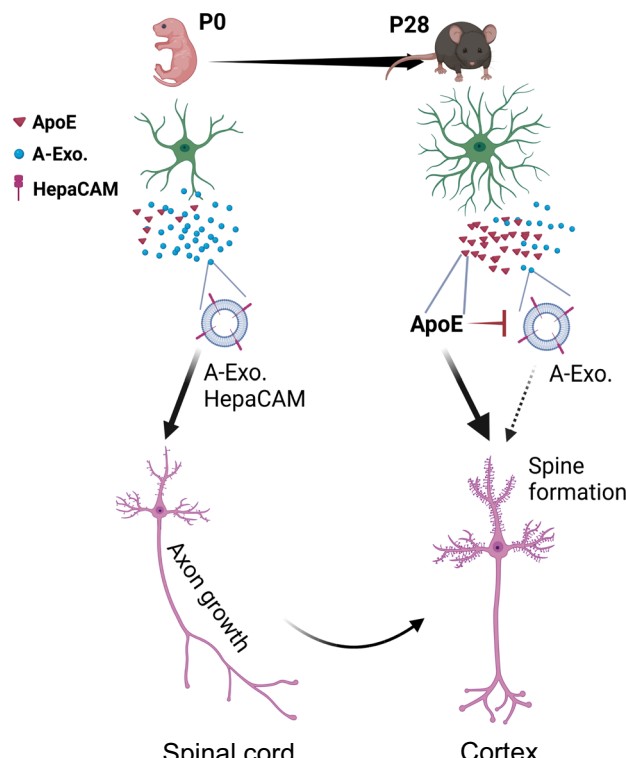

**Fig. 8 | Developmental astroglial exosome HepaCAM signaling and ApoE coordinates postnatal cortical pyramidal neuronal axon growth and dendritic spine formation.** Abundantly secreted astroglial exosomes promote CST axon growth during early postnatal development (within 1st postnatal week) when ApoE is lowly expressed; this effect is antagonized by increased ApoE expression to promote dendritic spine formation after CST axon growth is completed later during the postnatal development.

How ApoE inhibits the stimulatory effect of A-Exo. on axon growth remains unclear. Although HepaCAM is essential to mediate A-Exo's stimulation on axon growth, we found no evidence that ApoE binds to HepaCAM to block its stimulation on axon growth. In addition, ApoE can be readily separated from A-Exo. using the SEC with simple PBS wash, suggesting a non-covalent nature in the interaction between ApoE and A-Exo. While it is possible that other proteins on A-Exo. may mediate ApoE's inhibitory effect on A-Exo., ApoE has high affinity to cholesterol and phospholipids that are well distributed on A-Exo. surface. Thus, ApoE's direct binding to these lipids on A-Exo. may potentially mask HepaCAM's ECD or influence HepaCAM conformation to block A-Exo's surface contact with neurons especially growth cones. These potential mechanisms will be tested in future studies.

## Methods
This current study primarily uses mouse as the experimental model Care and treatment of animals in all procedures strictly followed the NIH Guide for the Care and Use of Laboratory Animals and the Guidelines for the Use of Animals in Neuroscience Research. Animal protocols used in this study have been approved (B2022-50) by the Tufts University IACUC.

### Reagents and neuronal culture treatments
Dynasore (Sigma-Aldrich), RNase A (Roche), Proteinase K (Fisher Scientific), CM-DiI (Thermo Fisher Scientific, C7001), and human Hepa-CAM protein extracellular domain (ECD, amino acid sequence 1-240, 16047-H08H) (Sino Biological Inc.) were used in this study. Dynasore (stock 50 mM) was prepared in DMSO and diluted 1000x in neuronal growth medium for treatment. Antibodies (final concentration 100 µg/ml) were mixed with neuronal growth medium and added onto primary neuronal cultures 2 h before A-Exo. treatment with exosomes. HepaCAM ECD coating is described below. Neuronal treatment with various drugs and/or exosomes was generally at DIV 3–4 for 24 h unless specifically described in main text.

### Mice
hCD63-GFP floxed mice were generated in the lab by homologous recombination, as previously described[35]. The WT (C57B/6J, #000664), Ai14-tdT^f/f reporter (#007914), Tg(Slc1a3-cre/ERT)1Nat/J (#012586), ApoE-KO (B6.129P2-Apoe tm1Unc/J #002052), B6.Cg-Tg (Thy1-YFP)HJrs/J (#003782), and B6.C-Tg(CMV-Cre)1Cgn/J(#006054) mice were obtained from the Jackson Laboratory. HepaCAM knock-out (KO) mice were generated by breeding HepaCAM floxed mice (a kind gift from Dr. Cagla Eroglu at Duke University)[19] with CMV-Cre mice. Both male and female mice were used in all experiments. Both sexes were used in all studies. All mice were maintained on a 12 h light/dark cycle with food and water ad libitum.

### Primary cortical astrocyte and neuronal culture
For cortical astrocyte cultures, P0-P3 mouse pups were decapitated, and cerebral cortices were removed and transferred into astrocyte growth medium (Dulbecco Minimum Essential Medium, DMEM, supplemented with 10% exosome-depleted FBS (fetal bovine serum, Gibco) and 1% penicillin/streptomycin) for dissection on ice. Meninges were stripped and cortices were minced and placed into 0.05% trypsin-EDTA solution for 10 min in a 37 °C water bath. The enzymatic reaction was stopped by addition of astrocyte culture medium. The tissue was washed twice with astrocyte medium and then gently dissociated by trituration with a fire-polished Pasteur pipette. Dissociated cells were filtered through a 70 µm strainer to collect a clear astrocyte cell suspension. Spinal cord astrocytes were cultured similarly as cortical astrocytes. Primary cortical neuron cultures were prepared from embryonic day 14–16 mouse brains. In brief, cortices were dissected and dissociated using 0.05% trypsin-EDTA solution for 10 min at 37 °C. Cells were seeded ($1 - 2 \times 10^4$/well) on Poly-D-lysine coated coverslips (Neuvitro), Poly-D-lysine and laminin coated coverslips (GG-12-Laminin Neuvitro) in 24-well culture dish ($1 - 2 \times 10^4$ cells/well) with 1 ml neuron plating medium containing DMEM, 10% FBS and 1% Pen-Strep at 37 °C in a humidified chamber of 95% air and 5% $CO_2$. After a 12 h seeding period, neuron plating medium was replaced by 700 µl neuron culture medium composed of neurobasal medium (Invitrogen), 2% B27 supplement (Thermo Fisher Scientific), 1% 100x GlutaMAX (Thermo Fisher Scientific), and 1% penicillin–streptomycin (Thermo Fisher Scientific). As we only observe <5% astrocytes in neuronal cultures and neuronal cultures were collected by DIV 7–8 at the latest, mitotic inhibitors such as cytosine arabinoside (Ara-C) were not used in neuronal cultures. DIV 4 neurons were usually used in added A-Exo. treatment.

### Intracellular cholesterol staining and quantification in primary astrocyte cultures
Intracellular cholesterol levels were measured with the cell-based cholesterol assay kit (Abcam, ab133116) Briefly, primary culture astrocytes were fixed with 4% PFA for 10 min. Astrocytes were stained with Filipin 3 according to manufacturer instructions. GFAP (rabbit anti-GFAP, 1:1000, Dako) immunostaining was also performed in primary astrocyte cultures (secondary antibody: anti-rabbit Alexa Fluor 555, 1:1000). Both GFAP immunostaining and Filipin 3 (with DAPI filter) signals were captured with the Zeiss Axio microscope (Zeiss, Heidelberg, Germany) using a ×20 objective lens. Filipin 3 signals within individual GFAP+ astrocyte that is outlined in ImageJ–FIJI was measured.

### Stereotaxic injections of AAV
For mouse (P60-90) spinal cord injections, AAV5-mCherry-*Gfap*-Cre virus was obtained from the University of North Carolina Vector Core (Chapel Hill, NC). Spinal cord ventral horn injections were performed

with a Hamilton Neuros Syringe with 33 G, point style 4, 45-degree bevel needle on a stereotaxic apparatus (Stoelting). A single dose of AAV5-mCherry-*Gfap*-Cre (0.5 µl, $4 \times 10^{12}$ genome copy (gc)/ml) was injected into the L1 segment of hCD63-GFP$^{f/+}$ or hCD63-GFP$^{f/+}$Ai14-tdT$^{f/+}$ mice, posterior to the median sulcus 0.4 mm laterally, 1.4 mm deep. Injections were performed at a rate of 0.1 µl/min. Post-operative care included injections of buprenorphine according to the IACUC requirement. For mouse pups (P1) motor cortex injections, hCD63-GFP$^{f/+}$Ai14-tdT$^{f/+}$ mouse pups were anaesthetized on ice for 3 min and then placed in a stereotaxic frame. AAV5-mCherry-G*fap*-Cre (0.3 µl, $4 \times 10^{12}$ gc/ml) was stereotaxically injected into the right side of motor cortex ($x = 1.0$ mm, $y = 1.8$ mm, $z = 0.6$ mm) using a 33-gauge needle. Injections were performed at a rate of 0.1 µl/min.

### Exosome purification and qNano particle analysis

Exosomes were prepared from astrocyte conditioned medium (ACM) from primary astrocyte culture (initial seeding: $4 \times 10^6$ cells/10 cm dish). After astrocytes become >90% confluent, the normal astrocyte growth medium was replaced with exosome depleted astrocyte growth medium composed of DMEM, 10% exosome-depleted FBS (Thermo Fisher Scientific), and 1% penicillin/streptomycin. ACM was replaced and collected every 3 days for up to 4 times (10 ml/10 cm dish). ACM was first spun at $300 \times g$ for 10 min at room temperature to remove suspension cells, then at $2000 \times g$ for 10 min at 4 °C to remove cell debris, then underwent following purification steps or stored at −80 °C. For ultracentrifugation (UC)-based purification, ACM was centrifuged at $10,000 \times g$ for 60 min at 4 °C. The supernatant was passed through a 0.22 µm polyether sulfone (PES) filter (Merck Millipore, MA, USA) followed by ultracentrifugation at $100,000 \times g$ for 60 min at 4 °C (SW 41 Ti Rotor, Beckman Coulter Inc). For size-exclusion chromatography (SEC) based isolation, ACM supernatant was first concentrated (to 500 µl) by centrifugation at $3500 \times g$ for 30 min at 4 °C using Centricon® Plus-70 Centrifugal Filter Devices with a 10k molecular weight cutoff (MilliporeSigma). Then the concentrated supernatant was passed through a 0.22 µm PES filter. The qEV original 35 nm columns (Izon Science, MA, USA) were then used according to the instructions of the manufacturer. Briefly, the column was rinsed with filtered PBS, and then 500 µl of concentrated and filtered supernatant from ACM was layered onto the top and each eluted fraction (500 µl/fraction) was collected. The eluted fractions were combined, as indicated in text and figure legend, and further concentrated using the Amicon Ultra-4 Centrifugal Filter Units (MilliporeSigma) in certain experiments. Tunable resistive pulse sensing (TRPS) by qNano particle analyzer (Izon Science, MA, USA) was used to measure the size distribution and quantity of isolated exosomes. Fifteen µl of concentrated and filtered ACM (500 µl from 10 ml/sample) or calibration particles included in the reagent kit were placed in the Nanopore (NP150, Izon Science). Samples were measured at 44 ~ 45 mm stretch with a voltage of 0.6 - 0.8 V at 1-pressure levels of 10 mbar. Particles were detected in short pulses of the current (blockades). The calibration particles were measured directly after the experimental sample under identical conditions. The data was processed using the Izon software (version 3.2).

### Exosome and HepaCAM coating for neuronal cultures

Sterile PDL or PDL/LN coated coverslips (Neuvitro) were rinsed twice with 1xPBS, then astroglial exosomes (1 µg) purified from 10 ml ACM were evenly added onto the top of coverslips and incubated for 1 h in a 37 °C cell culture incubator. Coverslips were then washed twice with 1x PBS before use. To block HepaCAM on exosome surface, 100 µg/ml of HepaCAM antibody (ProteinTech) was added separately on top of exosome coated coverslips and incubated for 1 h at 37 °C, then washed twice with 1x PBS before use. For HepaCAM-ECD coating, HepaCAM-ECD protein was diluted with PDL solution to 50 µg/ml, then 80 µl PDL/HepaCAM-ECD solution was added onto sterilized coverslips and

incubated for 2 h at room temperature. Coverslips were then washed twice with sterilized water before use.

### Biochemical treatment of exosomes

One µg A-Exo. (50 µl) was used in each treatment. For RNase treatment, RNase (Roche) was added to A-Exo. at a final concentration of 10 µg/ml for 5 min at 37 °C, then 20U SUPERase-In RNase inhibitor (Invitrogen) was added to block RNase activity. For proteinase K treatment, proteinase K was added to A-Exo at a final concentration of 10 µg/ml for 5 min at 37 °C, then 1% proteinase inhibitor cocktail (P8340, Sigma-Aldrich) was added. For treatment involving sonication, A-Exo. were sonicated at 50 Hz for 30 s on ice before RNase or proteinase K treatment. The reaction was washed two times using Amicon Ultra Centrifugal Filters (30 K MWCO, EMD Millipore) with 1x PBS to remove lysates. A final volume of 60 µl A-Exo. for the various treatments was then added to the primary neuronal culture.

### Immunocytochemistry, immunohistochemistry, live-cell, and confocal imaging

For immunocytochemistry, cultured neurons on cover slips were fixed in 4% paraformaldehyde for 15 min and permeabilized with 0.2% Triton X-100 for 5 min. The cells were blocked in 3% bovine serum albumin for 30 min and incubated with the following primary antibodies overnight at 4 °C: β-III tubulin (1:1000, MAB1195, R&D system), rabbit anti-MAP2 (1:1000, GeneTex, GTX133109), Gap43 Antibody (1:500, Novus Biologicals, clone 2G13), anti-mouse Tau (1:500, GeneTex), mouse anti-Map2 (1:1000, Sigma, M9942), rat anti-GFAP (1:5000, zymed, 273756), rabbit anti-GFAP (1:1000, Dako), and anti-human Tau (1:500, Dako). After incubation with the primary antibodies, neurons were washed three times with PBS, and incubated with following secondary antibodies for 1 h at room temperature: anti-mouse Alexa Fluor 488, anti-rabbit Alexa Fluor 568 and anti-goat 647 Alexa Fluor (1:1000, Invitrogen), and mounted with Prolong™ Glass Antifade Mountant with NucBlue™ Stain (Invitrogen).

For immunohistochemistry, mice were anaesthetized with a ketamine/xylazine cocktail and perfused with ice-cold PBS followed by ice-cold 4% paraformaldehyde. Dissected brains were post-fixed overnight in 4% paraformaldehyde at 4 °C for 24 h, and cryoprotected in 30% sucrose until tissue sinks. The tissue was embedded in OCT compound (Tissue-Tek) and 20 µm tissue sections were cut with a cryostat (Microm HM525). The following antibodies were used: GFAP (1:5000, Dako, #Z0334), mCD63 (1:50, MBL, #D263-3), and HepaCAM (1:200, R&D, #MAB4108). Primary antibodies were visualized with appropriate secondary antibodies conjugated with Alexa fluorophores (1:1000 Invitrogen) and mounted with Prolong™ Gold Antifade Mountant with DAPI (Invitrogen).

For imaging, cover slips were first grided into 16 (4 × 4) areas and images were taken from randomly distributed (both center and peripheral) areas for quantification. Low magnification (×20) images (βIII-tubulin/Map2 and spinal cord hCD63-GFP) were taken using the Zeiss Axio Imager fluorescence microscope, using the ZEN2 software to acquire and process the images. Confocal images of VGluT1/PSD95 and HepaCAM staining, hCD63-GFP, tdT, and Thy1-YFP fluorescence, were taken using the Leica SP8 FALCON confocal laser scanning microscope (15–20 µm Z stack with 0.5 µm step) magnified with ×63 (numerical aperture 1.0) objectives; images were processed with LAS X software.

For live-cell imaging and time-lapse, primary cortical neurons was performed on a Leica SP8 microscope 24 h following the addition of astroglial exosomes (1 µg). The microscope was equipped with a stage top incubator (model: INUBG2A-GSI2X TOKAI HIT) with temperature and $CO_2$ control to maintain an environment of 37 °C and 5% $CO_2$. The images were taken with a ×10 objective lens every 3 min for 8 h using the same exposure time. For axon growth rate quantification, primary neurons were plated onto cover slips coated with PDL, PDL + Hepacam-ECD, or PDL + A-Exo. On primary culture day 3, the live-cell

imaging was performed on a Leica SP8 microscope equipped with a stage top incubator to maintain a temperature of 37 °C and a $CO_2$ concentration of 5%. Images were captured every 5 min for 2 h using a ×20 objective lens with consistent exposure time. The resulting images were exported as multi-TIFF files and analyzed using Fiji ImageJ software (NIH). To measure axon growth rate, the longest neurite (considered to be the axon if longer than 50 µm) was traced using the SNT plugin in the image captured at time 0 min, and then traced again in the image captured at time 120 min. Growth rate was calculated as the increase in length divided by the 10 min time interval.

## Immunoblotting and immunoprecipitation

Mouse spinal cord, primary astrocyte pellets, and exosome fractions were homogenized with lysis buffer (Tris-HCL pH 7.4, 20 mM, NaCl 140 mM, EDTA 1 mM, SDS 0.1%, Triton X 1%, Glycerol 10%). Protease inhibitor cocktail (P8340, Sigma) and phosphatase inhibitor cocktail 3 (P0044, Sigma) was added in a 1/100 dilution to lysis buffer prior to tissue homogenization. Total protein amount was determined by DC™ Protein Assay Kit II (Bio-Rad), then lysates were loaded on 4–15% Mini-PROTEAN TGX Stain-Free Protein Gels (Bio-Rad). Separated proteins were transferred onto a PVDF membrane (Bio-Rad) with the Trans-Blot Turbo Transfer System (Bio-Rad). The membrane was blocked with 5% fat-free skim milk in TBST (Tris buffer saline with 0.05% Tween 20) or SUPERBLOCK T20 (TBS) Blocking Buffer (Thermo Fisher Scientific) then incubated with appropriate primary antibody overnight at 4 °C. The following primary antibodies were used: Thrombospondin (TSP)-1 (1:100, Santa Cruz clone SC-8), Thrombospondin (TSP)-2 (1:100, BD Biosciences), Hevin (1:200, R&D Systems), Sparc (1:200, R&D Systems), Sema3a (1:200, clone A-18 Santa Cruz), TSG101 (1:100, clone C-2 Santa Cruz), GFAP (1:2000, Dako, #Z0334), TurboGFP (1:2000, Evrogen, #AB513), mouse CD63 (1:200, MBL, # D263-3), CD81(1:1000, clone B-11, Santa Cruz), Alix (1:1000, Santa Cruz, clone 1A12, #sc-53540), GM130 (1:1000, Abclonal, #A5344), Calregulin (1:100, Santa Cruz, clone F-4 #sc-373863), HistoneH2A (1:400, Cell Signaling, #2718), GLAST (1:1000, GeneTex, #GTX134060), β-actin (1:1000, A1978, Sigma), HepaCAM N-terminal antibody (1:500, ProteinTech), HepaCAM C-terminal antibody (1:200, Affinity Biosciences, #DF12075), ApoE (1:1000, ABclonal, #A16344), ApoB (1:500, ABclonal, #A4184), ApoJ (1:500, ABclonal, #A1472). Secondary antibodies, including ECL anti-mouse IgG (1:10,000, GE HealthCare NA931V), anti-rabbit IgG-HRP (1:5000, GE Health Care NA934V), mouse anti-Goat IgG-HRP (1:1000, Santa Cruz) and anti-Rat IgG-HRP (1:5000, Thermo Fisher Scientific SC-2357) were diluted with Super Blocking Buffer. Bands were visualized on ChemiDoc MP imaging system (Bio-Rad) with ECL Plus chemiluminescent substrate (Thermo Fisher Scientific) or Clarity Max Western ECL Substrate (Bio-Rad). All uncropped and processed scans of immunoblots were included in the Source Data file in the Supplementary Information.

For immunoprecipitation, Dynabeads® M-270 Epoxy beads (Thermo Fisher Scientific) with anti-CD81 (clone Eat-2, BioLegend), anti-HepaCAM (Affinity Biosciences, # DF12075), and mouse IgG1 (clone MG1-45 BioLegend) was conjugated individually according to the instructions of the Dynabeads Antibody Coupling Kit (Thermo Fisher Scientific). Dynabeads (0.5 mg) were mixed with each antibody (5–10 µg) and incubated overnight at 4 °C with gentle agitation. Beads were then washed with washing buffer and 1xPBS. Concentrated ACM (500 µl, from 20 ml/sample) or exosomes isolated from SEC (2 µg/sample) were added and incubated overnight at 4 °C with rotating. IP mixes were then placed on a magnetic rack, washed 3 times, and eluted with western blot lysis buffer.

## LC-MS/MS proteomics and data analysis

Three biological exosome samples (20 µg/sample) were separated on 4–15% mini-protein TGX precast protein gels (Bio-Rad) and subsequently stained with Coomassie Blue, then each sample lane was excised and digested with trypsin and spiked with 0.2 pmol of ADH peptides (YEAST Alcohol dehydrogenase 1) at the Mass Spectrometry Facility at the University of Massachusetts Medical School. The samples were then injected into Orbitrap Fusion Lumos Mass Spectrometer (Thermo Fisher Scientific) in technical triplicates for label-free quantitation (LFQ) analysis. The data were searched against Swiss-Prot Mouse protein database using Mascot search engine through Proteome Discoverer software. The data was exported and normalized as intensity-based absolute quantification (iBAQ) quantitative values in Scaffold (version Scaffold_4.10, Proteome software). The selected parameters for protein identification were the following: Protein Threshold >95%; minimum 3 peptides per candidate protein; Peptide Threshold >90%; >1 × 10^5 iBAQ value in at least one of samples. The iBAQ value of the housekeeping protein ADH was used for normalization of biological replicates.

## Immuno-electron microscope (EM) imaging

EM imaging was performed in the Harvard Medical School Electron Microscopy Facility. AAV5-mCherry-*Gfap*-Cre injected hCD63-GFP^f/+ mice were perfused with 4% paraformaldehyde (PFA) and 0.1% glutaraldehyde. The spinal cord tissue was dissected out and post-fixed in 4% PFA for overnight, then spinal cord slices (100–200 µm) were prepared using vibratome and floated in PBS + 0.02 M glycine for 15 min. The slices were quenched, permeabilized, and blocked with blocking buffer (1% bovine serum albumin, 0.1% Triton X-100) at 4 °C. Anti-human CD63 (BD Pharmingen, #556019) antibody was then added and incubated overnight at 4 °C. The slices were washed three times for 20 min in PBS. The slices were incubated with Protein A-gold 5 nm (1:50, Utrecht, the Netherlands) for 1 h at 25 °C, washed in PBS and fixed in 1% (v/v) glutaraldehyde in PBS for 30 min. For Epon embedding, slices were incubated in 0.5% (w/v) osmium in ddH2O for 30 min, washed three times in ddH2O and then stepwise dehydrated (each step for 10 min) in 70% (v/v) ethanol, 95% (v/v) ethanol, and two times in 100% (v/v) ethanol. The slices were incubated in propyleneoxide, infiltrated in 50/50 propylenoxide/TAAB Epon, embedded in fresh TAAB Epon (Marivac Canada Inc) and polymerized at 60 °C for 48 h. The block was cut into 60 nm ultrathin sections using a Reichert Ultracut-S microtome. The slices were picked up on to copper grids that have been stained with uranyl acetate and lead citrate. Samples were examined using a JEOL 1200EX transmission electron microscope. Images were recorded with an AMT 2k CCD camera at ×30,000 magnification. For eluted fractions from SEC columns, negative staining was performed. Briefly, 5 µl of the sample was adsorbed to a carbon coated grid that had been made hydrophilic. The primary antibody used was anti-human CD63 (1:20, BD Pharmingen 556019), then samples were incubated with rabbit anti rat bridging antibody (1:50, Abcam ab6703) and Protein A-gold 10 nm (University Medical Center Utrecht, the Netherlands). Excess liquid was removed with filter paper (Whatman #1) and the samples were stained with 1% uranyl acetate. The grids were examined in a JEOL 1200EX transmission electron microscope and images were recorded with an AMT 2k CCD camera.

## Image analysis

For neurite tracing and Sholl analysis, healthy neurons with evidently elongated (>100 µm) neurites (βIII-tubulin⁺) or axons (βIII-tubulin⁺Map2⁻) from grided areas on cover slips from all groups were selected for quantification. Neurites and axons were traced and then measured using the Simple Neurite Tracer (SNT) plugin in Fiji ImageJ. Axons were defined as β-III tubulin⁺Map2⁻ neurites. Axon growth cone size was determined in Fiji ImageJ by manually tracing and measuring the area of regions of interest (ROIs) based on the anti-GAP43 antibody fluorescence at the tip of β-III tubulin⁺ (or Tau⁺) Map2⁻ axons. For quantification of VGluT1 or PSD95 puncta, confocal images were taken using the Leica SPE confocal laser scanning microscope (9–12 µm Z-stack with 0.5 µm step) magnified with ×63 objective and first converted to projection images (with maximal projection) for analyses. The software SynPAnal was used for quantifying the puncta density

and intensity/area of PSD95⁺ and VGLUT1⁺ puncta. Neurite segments (20–30 μm in length) were quantified from each neuron and their average values were also measured using SynPAnal software. Synapses were quantified, in a double-blind manner, based on the individual co-localization of PSD95 and VGLUT1, identified by the merge of their fluorescence along the selected segments. Multiple segments were quantified and averaged per neuron.

For extracellular and intracellular hCD63-GFP⁺ puncta analysis, the extracellular percentage ratio of hCD63-GFP⁺ puncta were determined in relation to the tdT⁺ astroglia using Fiji ImageJ based on confocal images. The hCD63-GFP channel image was first thresholded to create a binary black and red image. Then the Measure Analyzer tool was used to count all hCD63-GFP⁺ puncta area. The tdT channel image was thresholded and the Particle Analyzer tool was used to generate the ROIs of all tdT⁺ signals. Then the ROIs of tdT⁺ signals were overlaid on the hCD63-GFP⁺ images. hCD63-GFP⁺ area was then measured inside of tdT based ROIs. hCD63-GFP⁺ puncta inside tdT⁺ ROIs were considered as intracellular CD63-GFP⁺ signals. Extracellular hCD63-GFP⁺ area was determined by subtracting hCD63-GFP⁺ intracellular area from total hCD63-GFP⁺ area and the extracellular percentage ratio was calculated by dividing the total hCD63-GFP⁺ area by the extracellular hCD63-GFP⁺ area. 3D reconstruction of hCD63-GFP⁺ puncta and tdT⁺ astroglia at P6-8 and P28 was built using Surface function of the Imaris image analysis software (Bitplane) based on original Lecia confocal Z-stack files.

For quantification of dendritic spine density, confocal images of YFP⁺ pyramidal neurons of layer V motor cortex of Thy1-YFP⁺ and Thy1-YFP⁺ApoE⁻/⁻ mice were acquired at 0.5 μm intervals with a 63×oil immersion lens with Leica falcon confocal microscope. 3D reconstruction of YFP⁺ neurons was built using the Imaris image analysis software (Bitplane). Both apical collateral and basal dendrites and spines were traced with the filament tracing function in Imaris and quantified. The dendritic spine density was calculated by dividing the number of spines by dendrite length (-30–40 μm). In vivo 3D neuronal Sholl analysis was performed on basal dendrites with 10 μm increment radius following tracing Thy1-YFP⁺ pyramidal neurons in layer V of motor cortex with the SNT plugin in ImageJ.

### In vivo anterograde labeling of CST axons and measurement of spinal cord CST axon length

CST axons were anterogradely labeled by a single injection of the CM-DiI dye (10 mg/ml in N, N-dimethylformamide) into the right-side motor cortex of P1 pups with the use of Hamilton micro syringe with 33 gage 30° needle. Pups were perfused at P3 with cold 1x PBS, brains with spinal cord were fixed in 4% PFA overnight, and 100 μm sagittal cryosections were prepared along the anterior-posterior axis. They were mounted with Fluorogold anti-fade mounting medium then imaged under Keyence fluorescence microscope BZ-X700 with a Cy3 filter. Spinal cord CST axon length was measured based on the CM-DiI fluorescence signals from the superimposed images of individual mice (as shown in Fig. 5c) after the pyramidal decussation (PD) by using the segment line tool in ImageJ.

### In vivo exosomes isolation from CNS tissues

Brain and spinal cord tissue from P3-5 or P10 mice were used for isolation of CNS-derived exosomes. For P3-5, 3 pups were pooled as one sample for isolation. Following ketamine/xylazine overdose (i.p.), mice were perfused with ice-cold PBS, brain and spinal cord were rapidly dissected and placed in a 15 ml centrifuge tube, then processed for downstream analysis. After addition of 2 ml enzyme mix: 2 mg/ml collagenase D (Sigma-Aldrich) + 40 U/ml DNase I (Fisher Scientific, NC9709009) in HBSS- buffer, tissue was roughly homogenized with a 1 ml pipette tip and incubated for 30 min at 37 °C, aspirating every 5 min. Twenty μl proteinase inhibitor (P8340 Sigma-Aldrich) was added and the sample filtered used a 70 μm strainer. After serial centrifugation (10 min, 300 × g, RT; 10 min, 2000 × g, 4 °C; 20 min,

4500 × g, 4 °C), the final supernatant was passed through a 0.22 μm filter prior to fractionation with qEV 35 columns (Izon). Then collected #7–9 exosomes fractions (500 μl for each fraction) were concentrated with Amicon Ultra-4 30 k Centrifugal Filter (30 min, 4000 × g, 4 °C) and resuspended up to 100 ml PBS before storing at −80 °C.

### Statistical analysis

All statistical analyses were performed and graphs were generated using GraphPad Prism 8 and 9. Group differences in each assay at each time point were analyzed by two-tailed $t$-test (2 group comparison), one-way ANOVA (3 or more group comparison, 1 independent variable), or two-way ANOVA (3 or more group comparison, 2 independent variables). Statistical test(s) used are specified in figure legends. Data are presented as mean ± SEM unless otherwise described. No custom code was used in the analysis. Statistical significance was tested at a 95% ($p < 0.05$) confidence level and $p$ values are shown in each graph.

### Reporting summary

Further information on research design is available in the Nature Portfolio Reporting Summary linked to this article.

## Data availability

All data supporting this study are available upon request. Proteomic data of A-Exo. are available via ProteomeXchange with identifier PXD040650. Source data are provided with this paper.

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

## Acknowledgements

We thank Dr. Peter Juo (Tufts University School of Medicine), Dr. Fen-Biao Gao (University of Massachusetts Chan School of medicine), and Dr. Zhigang He (Boston Children's Hospital) for constructive discussions.

We thank Dr. Cagla Eroglu and Dr. Katherine Baldwin (Department of Cell Biology, Duke University School of Medicine) for providing HepaCAM floxed mice. This work was supported by NIH grants RF1AG057882, RF1AG059610, R01NS118747, R01NS125490, and R01AG078728 (Y.Y.). Imaging was performed with the assistance of the Tufts Center for Neuroscience Research. EM was performed with the assistance of the Harvard Medical School EM Core Facility. LC/MS/MS and proteomic analysis was performed with the help of University of Massachusetts Medical School.

## Author contributions

S.J. designed and performed the majority of experiments in this study and wrote part of the manuscript. Y.T. performed spinal cord injections and hCD63-GFP image analysis. X.C. performed immunostaining, image analysis, and wrote the manuscript. R.J. and V.P. performed image analysis, helped with exosome isolation, and wrote the manuscript. Y.Y. designed overall study, analyzed data, and wrote the manuscript.

## Competing interests

The authors declare no competing interests.
