## [Peer Review File · Nature Communications]

Astroglial exosome HepaCAM signaling and ApoE antagonization coordinates early postnatal cortical pyramidal neuronal axon growth and dendritic spine formationREVIEWER COMMENTS

Reviewer #1 (Remarks to the Author):

This manuscript discusses a very exciting and timely study on the function of astrocyte-derived extracellular vesicles in promoting synaptogenesis and axon growth. The authors convincingly show that astrocyte-derived EVs stimulate neurite outgrowth and that this effect is mediated by EV-associated HepaCAM. They also show that expression of ApoE at later developmental stages antagonizes the effect of these EVs. There are several minor limitations to this study, however, the novelty and rigor of the experimental data is convincing and limitations should be discussed as stated in the reviewer's critiques.

Critiques

1. Several studies have shown that GFAP and Glast1 are present in astrocyte-derived EVs. While the authors utilize GFAP-Cre driven expression of CD63-GFP labeled EVs from astrocytes, it is not clear if GFAP (or Glast1) as astrocytic marker is also present in these EVs. The authors should comment on whether these astrocytic markers are present in their EV preparations.
2. The authors use anti-CD81 and anti-Hepacam pulldown to convincingly demonstrate that this astrocytic adhesion protein is instrumental in the function of astrocyte-derived EVs for neurite outgrowth. However, it is not clear how large the proportion of HepaCAM EVs within the total population of astrocytic EVs is. It is hard to imagine that all of the astrocyte-derived EVs stimulate neurite outgrowth based on other publications showing that EVs from primary cultured astrocytes can have a variety of different effects on neurons. This should be discussed.
3. The in vivo experiments using AAV5-mediated expression of CD63-GFP convincingly show that astrocyte-derived EVs are taken up by neurons. This nicely aligns with the proposed model of developmentally regulated neuronal outgrowth and synaptogenesis by astrocyte-derived EVs. However, since HepaCAM is also a known expressed surface protein on astrocytes, it is not clear which of these effects is mediated by EVs and which are due to direct cell-to-cell contact between astrocytes and neurons. This limitation should be discussed.
4. The authors hypothesize that Hepacam EVs and ApoE secreted by astrocytes have developmentally regulated and antagonistic effects on axonogenesis vs. dendritogenesis, which is supported by the data obtained with the ApoE knockout mice. However, to really assess these developmental effects one would need to purify EVs from different developmental stages of the spinal cord and analyze their HepaCAM levels vs. ApoE in the non-EV fractions. As with critique point 3 and also 2, it is not clear which proportion of astrocytic EVs is critical at different developmental states in vivo and how this regulation compares to the effect of cell-to-cell contact between astrocytes and neurons. To clarify this question is probably difficult to answer since there are still no good loss-of-function in vivo models for EVs available. Nevertheless, the authors should discuss this limitation.

Reviewer #2 (Remarks to the Author):

In this manuscript the authors set out to determine the role of astrocyte secreted exosomes in axon formation. Using well controlled experiments, the authors isolated fractions from conditioned astrocyte media and showed the exosome enriched fractions can induce an increase in axon length of cultured primary neurons. They further show that this effect is mediated by HepaCAM, a transmembrane protein found on astrocyte secreted exosomes. Importantly, the authors show evidence that these processes occur in vivo, using various transgenic mouse lines. In a second line

of investigation, the authors show that ApoE secreted from astrocytes in non-exosome fractions inhibits the axon effects of astrocyte exosomes. They also show that lack of ApoE reduces dendritic spine formation and dendrite complexity.

There are two main strengths of this manuscript: 1. Careful isolation and characterization of extracellular vesicles/exosomes secreted by astrocytes. 2. Identification of effects on axons mediated by HepaCAM and astrocyte secreted extracellular vesicles/exosomes in vitro and in vivo. These studies show that astrocyte secreted exosomes/EVs can regulate axon formation through HepaCAM, which adds to the growing literature on the functional role of EVs in brain development.

However, the manuscript is weakened by the ApoE experiments in figures 6 and 7 as these are two independent lines of investigation. While it is interesting that ApoE can inhibit the effects of astrocyte exosomes, this opens the question of how relevant this is for the in vivo effects the authors observed. Indeed, the authors show that ApoE KO mice show normal axon lengths (7K). While the data on ApoE effects on dendrites is interesting, these data distract from the overall conclusions of the exosome studies and belong in another paper where more mechanistic insight can be included.

In general, the experiments were well carried out. Some minor critiques include:

1. The authors make the claim that they have isolated clean exosomes. However, in figure 1, the authors claim that SEC exosomes are cleaner as they are not positive for secreted proteins. Only 1 marker for exosomes (CD81) is used, whereas Ultracentrifugation (UC) based fractions have been stained for multiple markers (Supplementary Fig. 1a). They need to stain SEC fractions for other positive markers and a few negative markers (eg. Calnexin). It is also confusing why the authors show CD63 immunogold not but not western blot expression.
2. Are the EVs observed in fractions 7-8 really exosomes? While I applaud the careful characterization and immunodepleting of CD81 has effects, these experiments still do not definitely show that these EVs are derived from MVBs. The authors should add this caveat in the paper.
3. Data that uses primary cultures, biological replicates are mentioned in the figure legends. We assume this refers to using at least 2 different cultures (the authors should make this clear). However, it is not clear how many cells were quantified from each culture, and this should be shown in the figures so that culture variability can be assessed. Number of cells per groups is also rather low, it is usual practice to obtain at least 10 cells per culture.
4. In figure 2g, authors have demarcated the HepaCAM band around 50kDa as non-specific, but the one at 70kDa is its glycosylated form. However, in the manuscript, they describe the naive form of HepaCAM protein is predicted to be at ~50kDa. They need to clarify whether it is HepaCAM or non-specific.
5. Typo in line 268: "also undergoes a similar developmental up-regulation (Fig. 4c-d) as in cortex" It should be supplementary Fig. 4c-d.
6. While statistically significant, the low N in figure 4 (3-4 mice) for the AAV experiments may not be sufficient to make strong conclusions.
7. The mCherry expression in 4g is hard to see and should be quantified as in 4f to corroborate the author's conclusion that CD63 positive EVs can be found away from transduced neurons.
8. What fractions are used in the flow through experiments in Figure 6?

Reviewer #3 (Remarks to the Author):

In this manuscript by Jin et al. the authors examined the roles of astrocyte derived factors, HepaCAM on exosomes and secreted ApoE, on axon and dendrite development by cortical neurons. Using size exclusion chromatography (SEC) or ultracentrifugation (UC) of astrocyte conditioned media (ACM), the authors separated exosomes from secreted proteins and found that exosomes, but not soluble proteins, stimulated axon extension by cortical neurons. The axon outgrowth promoting effects of exosomes appear to be proteinaceous and not mRNA. Proteomic analysis showed 347 proteins identified on exosomes, including HepaCAM (37th most abundant). HepaCAM was subsequently identified within spinal cord cell lysates and on exosomes by blot. Exosome-derived HepaCAM is necessary to stimulate cortical axon outgrowth as exosomes isolated from HepaCAM KO animals does not stimulate cortical axon extension, and outgrowth can be inhibited with anti-HepaCAM antibodies. In fact, the extracellular domain alone coated as a substratum is sufficient to stimulate cortical axon extension. Next, the authors expressed CD63-GFP in astrocytes to track exosome distributions within and released from astrocytes. They find that immature astrocytes cultured from P8 pups have more robust secreted exosomes compared to P28 astrocytes. Interestingly, local infection of CD63-GFP of astrocytes within the spinal cord leads to robust spread of CD63-GFP signal 4 mm on either side of expressing astrocytes, suggesting that exosomes spread great distances in the spinal cord. To demonstrate a role of HepaCAM in vivo, the authors showed that CST axon extension into the spinal cord reduced in HepaCAM KO animals. In vitro, HepaCAM increases the size of growth cones, which was suggested to indicate faster axon extension. Next the authors tested whether ACM may contain other factors that modulate the outgrowth stimulating effects of exosomes. They found that ApoE contained within the soluble fractions reversed the growth promoting effects of exosomes, through an unknown mechanism. Finally, while ApoE blocks the outgrowth stimulating effects of exosomes, it appears to promote dendritic spines and synapse formation.

This is a largely well written paper that presents some interesting findings. The primary message that HepaCAM on astrocytes promotes axon outgrowth in vitro are the most convincing findings in my opinion. It is less clear to me the function of HepaCAM in vivo and the connection with ApoE. Below I outline by figure some of my concerns with this paper. Given the great many improvements I feel are necessary before publication in Nature Communications, it seems switching to a more specialized journal may be in the Authors best interest.

Figure 1: This figure is largely OK. I couple of small improvements would make data more convincing. First, I would like to see a second exosome marker on the blot in Fig 1B (maybe CD63?). Second, the representative images do not reflect well the quantitative measurements of axon lengths. For example, the average axon length in control images is nowhere near 500 um, which is what the graph indicates.

Figure 2: No issues.

Figure 3: Here the authors begin exploring the role of HepaCAM, which was one of 347 components identified on exosomes by mass spec. How did they come to select this component, which was much less abundant compared to many other proteins? While most proteins do not have clear roles in axon growth, some are known to influence axon extension (ie Tenascin-R). In any case, HepaCAM does appear to be the key component, as outgrowth stimulating activity is nearly completely lost in HepaCAM KO exosomes. It would be useful to show by blot that KO HepaCAM exosomes retain many of the other components.

Figure 4: This is perhaps the most compelling figure in this paper and therefore has the most questions. Viral infection to drive CD63-GFP and cytosolic mCherry in astrocytes is a powerful approach. Images show distinct puncta outside and apparently within astrocytes in vitro. Showing some orthogonal views (Y-Z) of confocal z stacks would verify CD63-GFP puncta are within astrocytes (may even use super resolution to improve z resolution). One important control would be to immunolabel CD63-GFP puncta for known exosome markers to confirm they are exosomes. In addition, staining them for HepaCAM would be interesting (in control and HepaCAM KO). Perhaps the most striking result from this paper is the extreme spread of exosomes from locally infected astrocytes within the spinal cord. It is hard for me to image this much signal being

generated from such a small region of mCh labelled astrocytes and how far exosomes can diffuse within dense spinal cord tissue. The authors sectioned the spinal cord, so it is difficult to assess the lateral (A-P) spread astrocytic processes from the infection point. If possible, wholemount confocal imaging of optically cleared spinal cords would allow 3D reconstruction of this tissue. This is a very robust and surprising finding, so I believe it is worth further exploration. Also, the CD63-GFP labeling looks cellular to me. Are these exosomes covering cells or have them been endocytosed by neighboring cells. Higher resolution imaging may resolve this. Panel d show immunolabeling for HepaCAM together with CD63-GFP. I am not sure the value of showing these together except to imply that exosomes are responsible for HepaCAM, but this does not show this. Instead, it would be very compelling to show that re-expressing HepaCAM regionally in astrocytes of the spinal cord of HepaCAM KO animals leads to deposition of HepaCAM widely across the spinal cord. This would most directly demonstrate what the authors are trying to imply with their approaches. They should also show HepaCAM IHC of a KO section. The final panels of figure 4 show immune EM images of CD63 labeling. However, it is not clear to me if these are from CD63-GFP expressing animals and sections were stained for GFP or is this native CD63 (doubtful as should be much more widespread). Also, it is not clear how they identified the astrocyte in their EM images.

Figure 5: This figure tries to address the role of HepaCAM in promoting axon extension in vivo. For this, the authors use a lipophilic dye to label CST motor neurons in wild-type and HepaCAM KO animals. While this is a useful approach and their quantification showed significant differences, I find the example images not very compelling and difficult to measure. Also, are HepaCAM KO animals normal? This seems counter to the implied role as key regulator of axon extension, but I guess redundancy in vivo may explain this. The remaining part of this figure measures growth cone size using GAP43 ICC, which the authors suggest indicates fast axon growth, which is not necessary true. In fact, some investigators have shown that large, highly lamellipodial growth cones are paused. It may be preferable to image F-actin, or better yet use live cell imaging to measure rates of axon extension +/- HepaCAM.

Figure 6: This figure switches to examine the modulatory role of ApoE, the rationale being that other investigators have shown that ACM contains ApoE, which promotes spines and synapse formation. They show that addition of ACM soluble fraction (FT), which contains ApoE, or purified ApoE, reverses the axon growth promoting effects of HepaCAM. The data here are the same as the assays shown in Figs 1 and 2. The mechanism how ApoE modulates the effects of HepaCAM are unknown and it is also not clear how this process may occur in vivo.

Figure 7: This figure shows that ApoE promotes spine and synapse formation, as has been published previously. The connection between the activity of HepaCAM on axon growth within the spinal cord, and potential activity of ApoE on dendrite development in the cortex is difficult to envision. The authors did test whether loss of ApoE altered CST axon growth in the spinal cord (would expect loss to improve growth if functions to suppress HepaCAM), but they saw no change.

Reviewer #4 (Remarks to the Author):

This manuscript reports a novel mechanism of stimulating axon outgrowth via astrocyte-derived exosomes. While several recent studies have focused on the roles of astrocyte-secreted factors in synapse formation, very little is known about the role of astrocyte-secreted factors in axon outgrowth. Therefore, this study is very exciting in concept as it addresses a striking knowledge gap within the field. Here the authors use an improved method of isolating astrocyte-secreted exosomes for in vitro study, as well as their previously published exosome reporter mouse, to demonstrate that astrocyte-exosomes promote axon outgrowth through a mechanism that requires hepCAM. Interestingly, the authors also find an inhibitory effect of APOE from non-exosome fractions on exosome-mediated axon outgrowth. Additionally, loss of APOE reduces dendritic spine formation. The findings from the paper have the potential to substantially advance

our understanding of how astrocytes regulate axon outgrowth and shed light on the function of astrocyte exosomes in the brain. However, in its current form, for many of the experiments, the sample sizes are too low and minimal explanation is given for inclusion and exclusion criteria for image acquisition and analysis. This raises concern with the robustness of the data that supports the main conclusions of the manuscript. Specific comments are listed below:

Specific comments:

1. For many of the cell culture experiments the number of biological replicates is too low. I will use Figure 1e as an example and then list at the end the other figure panels that have this same issue. In figure 1e, only 10 total neurons measured and from a total of 2 biological replicates, which does not provide convincing evidence of the reproducibility of this experiment. That only 10 neurons were quantified from two experiments raises concerns about the inclusion/exclusion criteria for imaging and quantifying neurite outgrowth. How were the regions of the dish/coverlip selected for imaging? (e.g., randomly, by someone blinded to the conditions?). Were all neurons in the image quantified or only neurons that had a neurite of a certain length? The inclusion/exclusion criteria for selecting neurons for imaging and analysis should be clearly detailed in the methods. Other figure panels with this same issue of insufficient number of biological replicates: Figure 1j; Figure 2b, d, e; Figure 3d, f; Figure 6b, f; Supplemental Figure 2d, h; Supplemental Figure 3c; Supplemental Figure 6b.
2. Please provide statistical analysis for supplemental Figure 2h
3. In Figure 3b, please provide P value for comparison of Control and Hepacam KO A-Exo
4. The P8 astrocytes in Figure 4b look extremely underdeveloped in comparison to other previously published studies that have characterized astrocyte morphology at this age (Bushong et al., 2004 and Stogsdill et al., 2017 are some examples). There are few branches visible and an atypical looking cell body. Little details about image acquisition are provided in the methods, so it is unclear if this is a single optical plane (which could explain the strange morphology) or a maximum projection image. The authors mentions that they did 3D renderings and measured volume. How many z-stacks were taken per astrocyte? It does not seem that they have taken enough to capture the entire astrocyte volume, so detailed methods need to be provided on image acquisition. They should be collecting the same amount of z-data for each image to keep measurements consistent.
5. There is no statistical information reported that accompanies the mentions in the text of the % if CD63-GFP+ puncta outside of TdT astroglia.
6. Does astrocyte exosome secretion change with astrocyte reactivity? Are any of the injection strategies, particularly in the P90 spinal cord, causing reactive gliosis?
7. Is it possible to use an alternate strategy that does not require injection? Such as the tamoxifen inducible Aldh1L1CreERT2, done at different time points and tissues collected a few days later?
8. For the in vitro synapse data in Figure 7 – is there any change in the co-localization of VGLUT1 and PSD95?
9. Related to #8, based on the information in the methods, the images that are analyzed for VGLUT1 and PSD95 puncta are max projection images of 9-12 μm z-stack. As projection image of this thickness is likely to cause puncta to overlap with each other in the z-direction, leading to false co-localization. For any co-localization studies, the authors should analyze projection images of $\sim 1 \mu\text{m}$

Reviewer #1 (Remarks to the Author):

This manuscript discusses a very exciting and timely study on the function of astrocyte-derived extracellular vesicles in promoting synaptogenesis and axon growth. The authors convincingly show that astrocyte-derived EVs stimulate neurite outgrowth and that this effect is mediated by EV-associated HepaCAM. They also show that expression of ApoE at later developmental stages antagonizes the effect of these EVs. There are several minor limitations to this study, however, the novelty and rigor of the experimental data is convincing and limitations should be discussed as stated in the reviewer's critiques.

Critiques

1. Several studies have shown that GFAP and Glast1 are present in astrocyte-derived EVs. While the authors utilize GFAP-Cre driven expression of CD63-GFP labeled EVs from astrocytes, it is not clear if GFAP (or Glast1) as astrocytic marker is also present in these EVs. The authors should comment on whether these astrocytic markers are present in their EV preparations.

We have now included new immunoblot results from A-Exo. samples to show that GFAP and several other typical intracellular organelles, cytoskeleton, and nuclear proteins are not detected on A-Exo., while GLAST (mostly monomers) are detected in A-Exo. in the revised Supplementary Fig. 1d and pages 5-6.

2. The authors use anti-CD81 and anti-Hepacam pulldown to convincingly demonstrate that this astrocytic adhesion protein is instrumental in the function of astrocyte-derived EVs for neurite outgrowth. However, it is not clear how large the proportion of HepaCAM EVs within the total population of astrocytic EVs is. It is hard to imagine that all of the astrocyte-derived EVs stimulate neurite outgrowth based on other publications showing that EVs from primary cultured astrocytes can have a variety of different effects on neurons. This should be discussed.

We have included additional discussion on potential astrocytic EV heterogeneity (page 22) in the revised manuscript. Although previous studies have shown that astrocytic EVs are able to have other effects on neurons, astrocytic EVs in these studies were all purified by the ultracentrifugation approach, which can be mixed with various astrocyte secreted proteins and elicits specific effects on neurons. Therefore, whether it is the astrocytic EV heterogeneity or other secreted proteins that differentially act on neurons needs to be further investigated in future studies. The percentage of HepaCAM+ exosomes in astrocytic EV population will also be investigated in future studies.

3. The in vivo experiments using AAV5-mediated expression of CD63-GFP convincingly show that astrocyte-derived EVs are taken up by neurons. This nicely aligns with the proposed model of developmentally regulated neuronal outgrowth and synaptogenesis by astrocyte-derived EVs. However, since HepaCAM is also a known expressed surface protein on astrocytes, it is not clear which of these effects is mediated by EVs and which are due to direct cell-to-cell contact between astrocytes and neurons. This limitation should be discussed.

We had briefly described that A-Exo. HepaCAM could "serve as an alternative cell to cell contact mechanism" in the original discussion (page 23), in recognizing that both modes of

action can be happening simultaneously. We have included additional discussion (page 23) in the revised manuscript.

As shown previously and in our results here, astrocyte morphology/processes are quite simple at early postnatal development which suggests a rather limited direct cell-to-cell contact interactions between astrocytes and neurons at that stage. In this case, secreted EVs can be important to mediate astroglial signaling and promote CST axon growth.

4. The authors hypothesize that Hepacam EVs and ApoE secreted by astrocytes have developmentally regulated and antagonistic effects on axonogenesis vs. dendritogenesis, which is supported by the data obtained with the ApoE knockout mice. However, to really assess these developmental effects one would need to purify EVs from different developmental stages of the spinal cord and analyze their HepaCAM levels vs. ApoE in the non-EV fractions. As with critique point 3 and also 2, it is not clear which proportion of astrocytic EVs is critical at different developmental states in vivo and how this regulation compares to the effect of cell-to-cell contact between astrocytes and neurons. To clarify this question is probably difficult to answer since there are still no good loss-of-function in vivo models for EVs available. Nevertheless, the authors should discuss this limitation.

Please see response above and we have included additional discussion (page 23) that both exosome and cell surface HepaCAM can co-exist and function to stimulate CST axon growth during early postnatal development.

While we agree that there are currently no good loss-of-function tools for in vivo EV yet, we have included new data to show that HepaCAM is clearly detected in EV fractions isolated from P3-5 CNS tissues (revised Fig. 4d) and ApoE is only detected from flow-through fractions from P10 CNS tissues (revised Supplementary Fig. 6b), which confirmed in vitro results and further supports the involvement of exosomal HepaCAM in promoting CST axon growth.

Reviewer #2 (Remarks to the Author):

In this manuscript the authors set out to determine the role of astrocyte secreted exosomes in axon formation. Using well controlled experiments, the authors isolated fractions from conditioned astrocyte media and showed the exosome enriched fractions can induce an increase in axon length of cultured primary neurons. They further show that this effect is mediated by HepaCAM, a transmembrane protein found on astrocyte secreted exosomes. Importantly, the authors show evidence that these processes occur in vivo, using various transgenic mouse lines. In a second line of investigation, the authors show that ApoE secreted from astrocytes in non-exosome fractions inhibits the axon effects of astrocyte exosomes. They also show that lack of ApoE reduces dendritic spine formation and dendrite complexity.

There are two main strengths of this manuscript: 1. Careful isolation and characterization of extracellular vesicles/exosomes secreted by astrocytes. 2. Identification of effects on axons mediated by HepaCAM and astrocyte secreted extracellular vesicles/exosomes in vitro and in vivo. These studies show that astrocyte secreted exosomes/EVs can regulate axon formation through HepaCAM, which adds to the growing literature on the functional role of EVs in brain development.

However, the manuscript is weakened by the ApoE experiments in figures 6 and 7 as these are two independent lines of investigation. While it is interesting that ApoE can inhibit the effects of astrocyte

exosomes, this opens the question of how relevant this is for the in vivo effects the authors observed. Indeed, the authors show that ApoE KO mice show normal axon lengths (7K). While the data on ApoE effects on dendrites is interesting, these data distract from the overall conclusions of the exosome studies and belong in another paper where more mechanistic insight can be included.

We disagree with the reviewer and actually think it is important and an integral part of the manuscript to include ApoE data, as both astrocyte exosomes and ApoE are regulating different developmental aspects of same population of cortical pyramidal (motor) neurons in the motor cortex layer V. This group of cortical motor neurons are special in that they extend long axons to spinal cord even at early postnatal days and then switch to dendritic spine development. Therefore, both astrocyte exosomes and ApoE are coordinated signaling to be involved in regulating these two processes of the same group of cortical motor neurons.

Although ApoE KO mice showed pretty normal CST axon length (Fig. 7k), this may not be surprising, as we showed that ApoE is developmentally up-regulated. ApoE has quite low expression in the first week (revised Supplementary Fig. 7d-e), mostly similar to ApoE KO condition.

In general, the experiments were well carried out. Some minor critiques include:

1. The authors make the claim that they have isolated clean exosomes. However, in figure 1, the authors claim that SEC exosomes are cleaner as they are not positive for secreted proteins. Only 1 marker for exosomes (CD81) is used, whereas Ultracentrifugation (UC) based fractions have been stained for multiple markers (Supplementary Fig. 1a). They need to stain SEC fractions for other positive markers and a few negative markers (eg. Calnexin). It is also confusing why the authors show CD63 immunogold not but not western blot expression.

We agree with the reviewer and have included additional immunoblot results with different markers to show that isolated exosomes express other typical exosome markers (CD63 and Alix, revised Fig. 1b) but not subcellular organelle, cytoskeleton, and nuclear markers (revised Supplementary Fig. 1d) to further support the efficient and selective isolation of astroglial exosomes using the SEC method.

2. Are the EVs observed in fractions 7-8 really exosomes? While I applaud the careful characterization and immunodepleting of CD81 has effects, these experiments still do not definitely show that these EVs are derived from MVBs. The authors should add this caveat in the paper.

As we applied 0.22um filtration before running the SEC column and exosomes are typical smaller EVs, we believe that our isolation favors exosomes (~50-200 nm) than larger size EVs (100-1000 nm). Whether these EVs are derived from MVBs will be investigated in future studies. We have discussed this point (pages 6 and 22) in the revised manuscript.

3. Data that uses primary cultures, biological replicates are mentioned in the figure legends. We assume this refers to using at least 2 different cultures (the authors should make this clear). However, it is not clear how many cells were quantified from each culture, and this should be shown in the figures so that culture variability can be assessed. Number of cells per groups is also rather low, it is usual practice to obtain at least 10 cells per culture.

We agree with the reviewer and have performed new experiments to increase the number of biological replicates in vitro and in vivo and included additional quantification for figures throughout the revised manuscript. We have also clarified the number of cells from each culture in the figure legend in the revised manuscript. Depending on the specific experiment, the number of cells per culture varies but all showed a clear and significant results with no ambiguity.

4. In figure 2g, authors have demarcated the HepaCAM band around 50kDa as non-specific, but the one at 70kDa is its glycosylated form. However, in the manuscript, they describe the naive form of HepaCAM protein is predicted to be at ~50kDa. They need to clarify whether it is HepaCAM or non-specific.

We have clarified that the size of detected HepaCAM is 70KDa band and the ~45KDa (more accurate estimate than the original ~50KDa) band is non-specific, as validated by HepaCAM KO samples. We have updated the text (page 9) accordingly in the revised manuscript.

5. Typo in line 268: “also undergoes a similar developmental up-regulation (Fig. 4c-d) as in cortex” It should be supplementary Fig. 4c-d.

We thank the reviewer for pointing out this error and have changed it in the revised manuscript.

6. While statistically significant, the low N in figure 4 (3-4 mice) for the AAV experiments may not be sufficient to make strong conclusions.

We have performed additional injections and quantifications to increase the number of mice per group (n = 6 mice/group now). The new group of mice were accidentally collected 2 days earlier at P6 (in green, revised Fig. 4c) instead of the original P8 (in blue, revised Fig. 4c) time point, which showed even more extracellularly localized hCD63-GFP+ exosomes. These two groups of mice at the early time point, together with P28 mice (also with increased number of mice), consistently shows observed developmental decrease of extracellularly localized astroglial exosomes.

7. The mCherry expression in 4g is hard to see and should be quantified as in 4f to corroborate the author’s conclusion that CD63 positive EVs can be found away from transduced neurons.

We have quantified the mCherry signals and have included in the revised Fig. 4f in the revised manuscript.

8. What fractions are used in the flow through experiments in Figure 6?

We used flow-through from A-Exo. purification which included all fractions from the SEC column in Fig. 6.

Reviewer #3 (Remarks to the Author):

In this manuscript by Jin et al. the authors examined the roles of astrocyte derived factors, HepaCAM on exosomes and secreted ApoE, on axon and dendrite development by cortical neurons. Using size exclusion chromatography (SEC) or ultracentrifugation (UC) of astrocyte conditioned media (ACM), the

authors separated exosomes from secreted proteins and found that exosomes, but not soluble proteins, stimulated axon extension by cortical neurons. The axon outgrowth promoting effects of exosomes appear to be proteinaceous and not mRNA. Proteomic analysis showed 347 proteins identified on exosomes, including HepaCAM (37th most abundant). HepaCAM was subsequently identified within spinal cord cell lysates and on exosomes by blot. Exosome-derived HepaCAM is necessary to stimulate cortical axon outgrowth as exosomes isolated from HepaCAM KO animals does not stimulate cortical axon extension, and outgrowth can be inhibited with anti-HepaCAM antibodies. In fact, the extracellular domain alone coated as a substratum is sufficient to stimulate cortical axon extension. Next, the authors expressed CD63-GFP in astrocytes to track exosome distributions within and released from astrocytes. They find that immature astrocytes cultured from P8 pups have more robust secreted exosomes compared to P28 astrocytes. Interestingly, local infection of CD63-GFP of astrocytes within the spinal cord leads to robust spread of CD63-GFP signal 4 mm on either side of expressing astrocytes, suggesting that exosomes spread great distances in the spinal cord. To demonstrate a role of HepaCAM in vivo, the authors showed that CST axon extension into the spinal cord reduced in HepaCAM KO animals. In vitro, HepaCAM increases the size of growth cones, which was suggested to indicate faster axon extension. Next the authors tested whether ACM may contain other factors that modulate the outgrowth stimulating effects of exosomes. They found that ApoE contained within the soluble fractions reversed the growth promoting effects of exosomes, through an unknown mechanism. Finally, while ApoE blocks the outgrowth stimulating effects of exosomes, it appears to promote dendritic spines and synapse formation.

This is a largely well written paper that presents some interesting findings. The primary message that HepaCAM on astrocytes promotes axon outgrowth in vitro are the most convincing findings in my opinion. It is less clear to me the function of HepaCAM in vivo and the connection with ApoE. Below I outline by figure some of my concerns with this paper. Given the great many improvements I feel are necessary before publication in Nature Communications, it seems switching to a more specialized journal may be in the Authors best interest.

Figure 1: This figure is largely OK. I couple of small improvements would make data more convincing. First, I would like to see a second exosome marker on the blot in Fig 1B (maybe CD63?). Second, the representative images do not reflect well the quantitative measurements of axon lengths. For example, the average axon length in control images is nowhere near 500 um, which is what the graph indicates.

We have included additional exosome markers CD63 and Alix in the revised Fig. 1b. We have updated representative images in Fig. 1d i.

Figure 2: No issues.

Figure 3: Here the authors begin exploring the role of HepaCAM, which was one of 347 components identified on exosomes by mass spec. How did they come to select this component, which was much less abundant compared to many other proteins? While most proteins do not have clear roles in axon growth, some are known to influence axon extension (ie Tenascin-R). In any case, HepaCAM does appear to be the key component, as outgrowth stimulating activity is nearly completely lost in HepaCAM KO exosomes. It would be useful to show by blot that KO HepaCAM exosomes retain many of the other components.

Cell adhesion protein has been known to regulate axon growth. Meanwhile, we have another study to investigate how inflammatory signals regulate astrocyte exosome composition and

function. Results from that study provided important insights to focus on HepaCAM. We are organizing these results to be published in a separate manuscript.

Indeed, we are also interested in knowing whether HepaCAM loss affects other proteins on astrocyte exosome or not. To get a full picture, we will be carrying out proteomic analysis on HepaCAM-deficient astrocyte exosomes to determine the overall changes of proteins in future studies.

Figure 4: This is perhaps the most compelling figure in this paper and therefore has the most questions. Viral infection to drive CD63-GFP and cytosolic mCherry in astrocytes is a powerful approach. Images show distinct puncta outside and apparently within astrocytes in vitro. Showing some orthogonal views (Y-Z) of confocal z stacks would verify CD63-GFP puncta are within astrocytes (may even use super resolution to improve z resolution). One important control would be to immunolabel CD63-GFP puncta for known exosome markers to confirm they are exosomes. In addition, staining them for HepaCAM would be interesting (in control and HepaCAM KO). Perhaps the most striking result from this paper is the extreme spread of exosomes from locally infected astrocytes within the spinal cord. It is hard for me to image this much signal being generated from such a small region of mCh labelled astrocytes and how far exosomes can diffuse within dense spinal cord tissue. The authors sectioned the spinal cord, so it is difficult to assess the lateral (A-P) spread astrocytic processes from the infection point. If possible, wholmount confocal imaging of optically cleared spinal cords would allow 3D reconstruction of this tissue. This is a very robust and surprising finding, so I believe it is worth further exploration. Also, the CD63-GFP labeling looks cellular to me. Are these exosomes covering cells or have them been endocytosed by neighboring cells. Higher resolution imaging may resolve this. Panel d show immunolabeling for HepaCAM together with CD63-GFP. I am not sure the value of showing these together except to imply that exosomes are responsible for HepaCAM, but this does not show this. Instead, it would be very compelling to show that re-expressing HepaCAM regionally in astrocytes of the spinal cord of HepaCAM KO animals leads to deposition of HepaCAM widely across the spinal cord. This would most directly demonstrate what the authors are trying to imply with their approaches. They should also show HepaCAM IHC of a KO section. The final panels of figure 4 show immune EM images of CD63 labeling. However, it is not clear to me if these are from CD63-GFP expressing animals and sections were stained for GFP or is this native CD63 (doubtful as should be much more widespread). Also, it is not clear how they identified the astrocyte in their EM images.

We thank the reviewer's constructive suggestions for this figure. We have included the representative orthogonal view of the confocal stacks of hCD63-GFP puncta in relation to tdT+ astrocytes (revised Supplementary Fig. 4b). We have also performed CD63 immunostaining and found well co-localized signals between CD63 immunoreactivity and hCD63-GFP signals (revised Supplementary Fig. 4g), confirming their exosome identity.

Indeed, we had performed HepaCAM immunostaining and found co-localized HepaCAM immunoreactivity on hCD63-GFP+ exosomes (revised Supplementary Fig. 4d). We have also included HepaCAM immunostaining on HepaCAM KO sections (had done it before but forgot to include in the original submission) with undetected HepaCAM immunoreactivity on these sections (revised Supplementary Fig. 4d), which confirms specific HepaCAM immunoreactivity. We don't think re-expressing HepaCAM onto the HepaCAM KO mice would be needed. This is also technically infeasible to express HepaCAM in the spinal cord at this early postnatal stage.

We have also included new data to show that HepaCAM is clearly detected in EV fractions isolated from P3-5 CNS tissues (revised Fig. 4d), further confirming the co-localization between HepaCAM immunoreactivity and hCD63-GFP (revised Supplementary Fig. 4d).

We have also quantified mCherry signal spreading (< 1000 μ m) and included in the revised Fig. 4f. The strong signals at the injection site suggest that our AAV injections went well which leads to sufficient hCD63-GFP expression. How exosome signals spread laterally is not studied here and will need other strategies to examine. Meanwhile, we performed new injections of AAV-Gfap-Cre into hCD63-GFP^{f/+} Ai14-tdT^{f/+} mice and acquired high magnified confocal images (revised Fig. 4g) to better visualize the intracellular and extracellular hCD63-GFP signals in relation to the labeled astrocytes. How exactly A-Exo. spread long distance in the spinal cord and be endocytosed by other cell types will be studied in the future.

We have clarified that the immunoEM was based on hCD63 antibody, thus signals representing induced hCD63 expression. Astrocytes were identified based on quite translucent cytoplasm, lack of synaptic features, and less polarized morphology.

Figure 5: This figure tries to address the role of HepaCAM in promoting axon extension in vivo. For this, the authors use a lipophilic dye to label CST motor neurons in wild-type and HepaCAM KO animals. While this is a useful approach and their quantification showed significant differences, I find the example images not very compelling and difficult to measure. Also, are HepaCAM KO animals normal? This seems counter to the implied role as key regulator of axon extension, but I guess redundancy in vivo may explain this. The remaining part of this figure measures growth cone size using GAP43 ICC, which the authors suggest indicates fast axon growth, which is not necessary true. In fact, some investigators have shown that large, highly lamellipodial growth cones are paused. It may be preferable to image F-actin, or better yet use live cell imaging to measure rates of axon extension +/- HepaCAM.

Our images with dye-labeled CST axons are typical with the dye-labeling approach especially with the clear pyramidal decussation (PD). As described in the text, other genetically based approaches to label CST axons do not work at this early postnatal stage.

HepaCAM KO mice appear normal without any obvious behaviors, weight, or breeding changes. This indeed could be due to compensatory or redundant mechanisms. HepaCAM KO A-Exo. still stimulates axon growth, but much more modestly (revised Fig. 3b), indicating additional signals/mechanisms can be involved.

We did not suggest that growth cone size promotes “fast” axon growth. We actually only described that the substrate plays important roles in growth cone morphology and axon growth. As described in the manuscript (page 16), “fan-shaped growth cone morphology with extended peripheral domain....., is a characteristic growth cone morphology induced by CAM substrates, but not LN substrates”, which supports the promoting role of HepaCAM in axon growth. Meanwhile, we had done the F-actin staining but it did not work very well to illustrate the growth cone.

Nonetheless, we have performed live cell imaging experiments to compare the axon growth rate between HepaCAM ECD and A-Exo. and found a very comparable axon growth rate (new Fig. 5i), further suggesting that HepaCAM ECD-induced growth cone size expansion is functional as that by A-Exo. but not paused.

Figure 6: This figure switches to examine the modulatory role of ApoE, the rational being that other

investigators have shown that ACM contains ApoE, which promotes spines and synapse formation. They show that addition of ACM soluble fraction (FT), which contains ApoE, or purified ApoE, reverses the axon growth promoting effects of HepaCAM. The data here are the same as the assays shown in Figs 1 and 2. The mechanism how ApoE modulates the effects of HepaCAM are unknown and it is also not clear how this process may occur in vivo.

We think that this is a very important result to specify that ApoE is mostly expressed in exosome-free fractions but not in astroglial exosomes. Based on that, we further demonstrated that ApoE strongly inhibits astroglial exosome effect in promoting axon growth. Although ApoE appears not interacting with HepaCAM directly, this interesting antagonizing effect of two kinds of secreted signals from astrocytes is a mechanistic dissection and advances our understanding how different astrocytes secreted signals interact with each other to affect neuronal development.

Meanwhile, we had discussed how ApoE may inhibit astroglial exosome effect on axon growth even not directly binding to HepaCAM in the last paragraph of the discussion (page 25).

Moreover, we have included new data to show that HepaCAM is clearly detected in EV fractions isolated from P3-5 CNS tissues (revised Fig. 4d) and ApoE is only detected from flow-through fractions from P10 CNS tissues (revised Supplementary Fig. 6b). We think their interaction depends on specific in vivo local context, secretion sites, and expression (quantity) levels, etc. We will continue to investigate that in future studies.

Figure 7: This figure shows that ApoE promotes spine and synapse formation, as has been published previously. The connection between the activity of HepaCAM on axon growth within the spinal cord, and potential activity of ApoE on dendrite development in the cortex is difficult to envision. The authors did test whether loss of ApoE altered CST axon growth in the spinal cord (would expect loss to improve growth if functions to suppress HepaCAM), but they saw no change.

While prior studies have shown the synapse formation role of ApoE/cholesterol, these studies were all carried out in vitro. Whether ApoE deficiency affects dendritic spine formation in vivo was not previously explored.

As described above, our results showed that astrocyte exosomes and ApoE regulate different developmental aspects of same population of cortical pyramidal (motor) neurons in the motor cortex layer V. This group of cortical motor neurons are special in that they extend long axons to spinal cord even at early postnatal days and then switch to dendritic spine development. Therefore, it is an integral signaling coordination between astrocyte exosomes and ApoE to be involved in regulating these two processes of the same group of cortical motor neurons.

We actually think it is not surprising that ApoE KO mice showed pretty normal CST axon length, as we showed that ApoE is developmentally up-regulated. ApoE has quite low expression in the first week (Supplementary Fig. 7), mostly similar to ApoE KO condition.

Reviewer #4 (Remarks to the Author):

This manuscript reports a novel mechanism of stimulating axon outgrowth via astrocyte-derived

exosomes. While several recent studies have focused on the roles of astrocyte-secreted factors in synapse formation, very little is known about the role of astrocyte-secreted factors in axon outgrowth. Therefore, this study is very exciting in concept as it addresses a striking knowledge gap within the field. Here the authors use an improved method of isolating astrocyte-secreted exosomes for in vitro study, as well as their previously published exosome reporter mouse, to demonstrate that astrocyte-exosomes promote axon outgrowth through a mechanism that requires hepaCAM. Interestingly, the authors also find an inhibitory effect of APOE from non-exosome fractions on exosome-mediated axon outgrowth. Additionally, loss of APOE reduces dendritic spine formation. The findings from the paper have the potential to substantially advance our understanding of how astrocytes regulate axon outgrowth and shed light on the function of astrocyte exosomes in the brain. However, in its current form, for many of the experiments, the sample sizes are too low and minimal explanation is given for inclusion and exclusion criteria for image acquisition and analysis. This raises concern with the robustness of the data that supports the main conclusions of the manuscript. Specific comments are listed below:

Specific comments:

1. For many of the cell culture experiments the number of biological replicates is too low. I will use Figure 1e as an example and then list at the end the other figure panels that have this same issue. In figure 1e, only 10 total neurons measured and from a total of 2 biological replicates, which does not provide convincing evidence of the reproducibility of this experiment. That only 10 neurons were quantified from two experiments raises concerns about the inclusion/exclusion criteria for imaging and quantifying neurite outgrowth. How were the regions of the dish/coverslip selected for imaging? (e.g., randomly, by someone blinded to the conditions?). Were all neurons in the image quantified or only neurons that had a neurite of a certain length? The inclusion/exclusion criteria for selecting neurons for imaging and analysis should be clearly detailed in the methods.

We thank the reviewer for pointing out the lack of specific information. We forgot to include this information during the initial submission. We have now included specific information about neuronal quantification in the materials and methods of the revised manuscript. Basically, coverslips were first grided into 16 (4x4) areas and images were taken from randomly distributed (both center and peripheral) areas for quantification. Neurons with at least 100 μ m neurite length were included in the quantification

We have also performed new experiments to increase biological replicates in vitro and in vivo and included additional quantification for figures throughout the revised manuscript.

Other figure panels with this same issue of insufficient number of biological replicates: Figure 1j; Figure 2b, d, e; Figure 3d, f; Figure 6b, f; Supplemental Figure 2d, h; Supplemental Figure 3c; Supplemental Figure 6b.

We have specifically increased the number of neurons from new experimental replicates and additional quantification specifically for the figures mentioned above. We would like to point out that the inclusion of additional data points further strengthened our observation and conclusion.

2. Please provide statistical analysis for supplemental Figure 2h

We have included statistical analysis in the revised Supplementary Fig. 2h.

3. In Figure 3b, please provide P value for comparison of Control and Hepacam KO A-Exo

We have included p values in the revised Fig. 3b for the comparison between control and HepaCAM KO A-Exo. on axon growth. As we initially described, HepCAM KO A-Exo. is still able to stimulate axon growth, but only modestly.

4. The P8 astrocytes in Figure 4b look extremely underdeveloped in comparison to other previously published studies that have characterized astrocyte morphology at this age (Bushong et al., 2004 and Stogsdill et al., 2017 are some examples). There are few branches visible and an atypical looking cell body. Little details about image acquisition are provided in the methods, so it is unclear if this is a single optical plane (which could explain the strange morphology) or a maximum projection image. The authors mentions that they did 3D renderings and measured volume. How many z-stacks were taken per astrocyte? It does not seem that they have taken enough to capture the entire astrocyte volume, so detailed methods need to be provided on image acquisition. They should be collecting the same amount of z-data for each image to keep measurements consistent.

We have now replaced the initial tdT+ astrocyte confocal image at P8 with a new tdT+ astrocyte confocal image. These representative images are maximum projected images but not from a single optical plane. For the 3D renderings, confocal image stacks with 15-20um with 0.5um step were collected which is sufficient to build astrocyte domains. Meanwhile, astrocytes at P6-8 age are typically harder to build as their morphology is still primitive at that stage.

5. There is no statistical information reported that accompanies the mentions in the text of the % of CD63-GFP+ puncta outside of TdT astroglia.

We have included this statistical p value in the text (page 12).

6. Does astrocyte exosome secretion change with astrocyte reactivity? Are any of the injection strategies, particularly in the P90 spinal cord, causing reactive gliosis?

We have a separate study to specifically examine how exosome secretion is altered from inflammatory signals-induced reactive astrocytes in vitro and in vivo. There are dramatic changes in exosome secretion in these reactive astrocytes. That study is progressing along and will be organized in a separate manuscript for publication in the near future.

The stereotaxic injection most likely causes reactive gliosis right after the injection, but this gliosis is likely to resolve as the damage is very focal and there is two weeks period before the mice were collected for analysis. It is also less relevant to our goal which is not to see how this injury affects exosome secretion. On the other hand, the focal injection strategy in spinal cord is needed to observe the spreading of labeled astroglial exosomes.

7. Is it possible to use an alternate strategy that does not require injection? Such as the tamoxifen inducible Aldh1L1CreERT2, done at different time points and tissues collected a few days later?

This is certainly feasible to use Aldh1l1-CreERT2 mice breeding. Indeed, we have used Slc1a2 (GLAST)-CreERT2 mice to breed with hCD63-GFP^{fl} mice (revised Supplementary Fig. 4c). However, this approach would not be able to show specific spreading of secreted astroglial

exosomes, as hCD63-GFP signals will be induced from large number of astrocytes in the whole CNS.

8. For the in vitro synapse data in Figure 7 – is there any change in the co-localization of VGluT1 and PSD95?

We have now included the co-localization analysis and found much reduced synapse density in neurons treated with ApoE KO ACM compared to WT ACM-treated neurons (revised Supplementary Fig. 7c). During this process, we realized that there were calculation errors in the original PSD95 density quantification following WT and ApoE KO ACM treatment. We have thoroughly checked all data and included the correct PSD95 density quantification in revised Fig. 7c.

9. Related to #8, based on the information in the methods, the images that are analyzed for VGluT1 and PSD95 puncta are max projection images of 9-12 μm z-stack. As projection image of this thickness is likely to cause puncta to overlap with each other in the z-direction, leading to false co-localization. For any co-localization studies, the authors should analyze projection images of $\sim 1 \mu\text{m}$

While we agree that maximum projection images can potentially have puncta overlapped in z-direction, given the low number of z sections from cultured neurons (unlike tissue sections) and low numbers of co-localization between VGluT1 and PSD95, we don't see dramatic differences in quantifying puncta based on maximum projected images vs. single layer. On the other hand, maximum projected images significantly facilitate the identification of puncta on neurites which helps achieve a more accurate quantification than single layer. Therefore, we presented co-localization quantification based on maximum projection images in the revised Supplementary Fig. 7c.

REVIEWER COMMENTS

Reviewer #1 (Remarks to the Author):

The authors have adequately responded to the reviewer's critique and revised the manuscript accordingly.

Reviewer #2 (Remarks to the Author):

The revised manuscript has mostly addressed my original critiques. The added info on stats/culture data etc makes the data much more convincing. I still have some corrections for the authors:

1. Many papers have now shown that classifying EVs purely by size is not useful. For example, not all EVs that are 50-200nm are exosomes.
2. I still find the addition of the ApoE data as distracting. While I understand the authors want to contrast non-EV signaling with astrocyte secreted EVs, this significance is not clear from the introduction or discussion. Reviewer 3 had similar issues with the connection to ApoE, but the authors seems adamant to include this data. In the end, this should preclude publication but some improvement in discussing the connection/significance should be attempted.

Reviewer #3 (Remarks to the Author):

In the revised manuscript by Jin et al. the authors did address several concerns that I listed, but several were not considered or were improperly addressed in my opinion. The changes I liked included using 3D rendering to convincingly demonstrate intracellular vs extracellular exosomes and live cell imaging to demonstrate faster axon extension rates on HepaCAM. However, there were a few additional changes I would have preferred to see. For example, in figure 4 I asked the authors to show that the CD63-GFP puncta were in fact labeling exosomes. To address this the authors immunolabeled for mouse CD63. While this will label endogenous mouse CD63, it is not clear that it will also label expressed hCD63-GFP? Even if this does label distinct populations of CD63, I believe it would be much more compelling to show that expressed hCD63-GFP overlaps with one (or more) different markers of exosomes. hCD63-GFP is a prominent tool used throughout the paper, so it seems to validate it as an accurate marker of exosomes would be important. They did show that HepaCAM immunoreactivity is around hCD63-GFP exosomes, but this is not surprising as endogenous HepaCAM will be deposited everywhere and independent of marked hCD63-GFP exosomes. However, it is unclear to me why hCD63-GFP exosomes would not also label strongly for HepaCAM if they are depositing it? Most importantly, in my opinion, the best way to demonstrate that HepaCAM is being deposited widely by local release specifically from astrocytes is to express

HepaCAM locally and exclusively within astrocytes within KO animals. This would directly inform how much HepaCAM comes from astrocytes and how far it spreads. This may require some sort of inducible expression HepaCAM mouse, which they may not have? They did add new immunolabeling for β III tubulin, but this staining for some reason looks nothing like axons within the spinal cord (Supp 4c). Finally, I still find the link between ApoE and HepaCAM a bit fuzzy. The authors suggest both are released from astrocytes (ApoE secreted and HepaCAM on exosomes), but HepaCAM modulates axon growth in the spinal cord, while ApoE regulates spine formation later in development in the cortex. It is not clear if these two factors ever interact in space and time as being suggested here.

Reviewer #4 (Remarks to the Author):

The authors have sufficiently addressed my concerns and I have no further concerns. The additional replicates and the detailed description of imaging and analysis workflows make the data more robust and strengthen the conclusions of the paper.

Reviewer #1 (Remarks to the Author):

The authors have adequately responded to the reviewer's critique and revised the manuscript accordingly.

Reviewer #2 (Remarks to the Author):

The revised manuscript has mostly addressed my original critiques. The added info on stats/culture data etc makes the data much more convincing. I still have some corrections for the authors:

1. Many papers have now shown that classifying EVs purely by size is not useful. For example, not all EVs that are 50-200nm are exosomes.

We agree with the reviewer on this and are not suggesting that exosomes and other EVs are simply separating by the size. We are developing new tools to be able to separate plasma membrane-derived vesicles vs. endosome-derived exosomes and will test these tools in future studies.

2. I still find the addition of the ApoE data as distracting. While I understand the authors want to contrast non-EV signaling with astrocyte secreted EVs, this significance is not clear from the introduction or discussion. Reviewer 3 had similar issues with the connection to ApoE, but the authors seems adamant to include this data. In the end, this should preclude publication but some improvement in discussing the connection/significance should be attempted.

Firstly, we included new data to show that ApoE can be detected in developing spinal cords as well (new Supple. Fig. 7h-i). Secondly, we had done experiments to show that ApoE inhibition on A-Exo. also occurs in spinal cord astrocyte exosomes (new Supple. Fig. 6c).

In addition, gradual increase of ApoE expression levels in first postnatal week at spinal cords (new Supple. Fig. 7h-i) is consistent with the observation that ApoE dose-dependently inhibits A-Exo effect on axon growth (Fig. 6d-e). Therefore, these results support the notion that as the development goes, gradually increased ApoE can be increasingly suppressive for A-Exo. effect on axon growth in the spinal cord. Meanwhile, cortically increased ApoE promotes spine formation on dendrites of same group layer V pyramidal neurons in motor cortex. We have included these discussions in the revised manuscript (pages 24-25).

After considering reviewer comment, we still think it is relevant to include these interesting results that ApoE inhibits A-Exo. effect on axon growth. This indeed allows contrast of non-EV and EV signaling from astrocytes, as reviewer commented.

Reviewer #3 (Remarks to the Author):

In the revised manuscript by Jin et al. the authors did address several concerns that I listed, but several were not considered or were improperly addressed in my opinion. The changes I liked included using 3D rendering to convincingly demonstrate intracellular vs extracellular exosomes and live cell imaging to demonstrate faster axon extension rates on HepaCAM. However, there were a few additional changes I would have preferred to see. For example, in figure 4 I asked the authors to show that the CD63-GFP

puncta were in fact labeling exosomes. To address this the authors immunolabeled for mouse CD63. While this will label endogenous mouse CD63, it is not clear that it will also label expressed hCD63-GFP? Even if this does label distinct populations of CD63, I believe it would be much more compelling to show that expressed hCD63-GFP overlaps with one (or more) different markers of exosomes. hCD63-GFP is a prominent tool used throughout the paper, so it seems to validate it as an accurate marker of exosomes would be important. They did show that HepaCAM immunoreactivity is around hCD63-GFP exosomes, but this is not surprising as endogenous HepaCAM will be deposited everywhere and independent of marked hCD63-GFP exosomes. However, it is unclear to me why hCD63-GFP exosomes would not also label strongly for HepaCAM if they are depositing it? Most importantly, in my opinion, the best way to demonstrate that HepaCAM is being deposited widely by local release specifically from astrocytes is to express HepaCAM locally and exclusively within astrocytes within KO animals. This would directly inform how much HepaCAM comes from astrocytes and how far it spreads. This may require some sort of inducible expression HepaCAM mouse, which they may not have? They did add new immunolabeling for β III tubulin, but this staining for some reason looks nothing like axons within the spinal cord (Supp 4c). Finally, I still find the link between ApoE and HepaCAM a bit fuzzy. The authors suggest both are released from astrocytes (ApoE secreted and HepaCAM on exosomes), but HepaCAM modulates axon growth in the spinal cord, while ApoE regulates spine formation later in development in the cortex. It is not clear if these two factors ever interact in space and time as being suggested here.

We thank reviewer comment and have attempted immunostaining of other known exosome markers Tsg101 and Alix. We used antibodies previously reported (for example, Ruan Z et al., Molecular Neurodegeneration, 2020). However, we did not observe obvious and specific immuno signals. When confocal images were acquired with increased gain, images are highly similar to 2nd antibody only (no primary antibody) images with irregular long process-looking signals (see Figure. 1). Therefore, we conclude that these signals are non-specific. On the other hand, we have performed immunostaining with another mouse-specific CD63 antibody (MBL, #D263-3) we and others previously validated and consistently found co-localization with hCD63-GFP without noticeable background (new Supple. Fig. 4h). It is possible that previously used mCD63 antibody also detects hCD63, thus we replace immunostaining results with this new mCD63 antibody in new Supple. Fig. 4h. In addition, we included new data to show that we can indeed detect GFP tag and exosome markers Alix/CD81 from A-Exo. collected from 4-OHT-treated Slc1a3-CreER⁺hCD63-GFP^{i/+} astrocyte cultures (new Supple. Fig. 4a), confirming the secretion of GFP-tagged A-Exo.

In previous revision, we have included HepaCAM immunoblot from in vivo isolated exosome fraction (Fig. 4d) to show that HepaCAM is indeed detected in the exosome fraction but not in the flow-through of brain and spinal cord tissues. Together with the HepaCAM staining results, we think it demonstrated that HepaCAM is indeed on A-Exo. It is not currently feasible to selectively induce HepaCAM expression in astrocytes and also have A-Exo. labeled to show that HepaCAM is on labeled A-Exo. in early postnatal stages in HepaCAM KO mice but we will think about how to develop new tools for that in future studies.

As described above, we included new data to show that ApoE can be detected in developing spinal cords as well (new Supple. Fig. 7h-i). In addition, we had done experiments to show that ApoE inhibition on A-Exo. also occurs with spinal cord astrocyte exosomes (new Supple. Fig. 6c).

These new results and also the observation that ApoE inhibition on A-Exo. is dose-dependent (Fig. 6d-e) support the notion that as the development goes, gradually increased ApoE can be increasingly suppressive for A-Exo. effect on axon growth in the spinal cord. Note that this may not be mediated through direct ApoE and HepaCAM interaction, as we discussed. On the other hand, cortically increased ApoE promotes spine formation on dendrites of cortical pyramidal neurons. Thus, ApoE may have region-dependent functions in regulating postnatal development of same group layer V pyramidal neurons, which we think is relevant to include. We have included these discussions in the revised manuscript (pages 24-25).

Reviewer #4 (Remarks to the Author):

The authors have sufficiently addressed my concerns and I have no further concerns. The additional replicates and the detailed description of imaging and analysis workflows make the data more robust and strengthen the conclusions of the paper.

REVIEWERS' COMMENTS

Reviewer #2 (Remarks to the Author):

I have nothing further to add to this review.

Reviewer #3 (Remarks to the Author):

I have no additional comments.